



# The extreme drought of 1842 in Europe as described by both documentary data and instrumental measurements

Rudolf Brázdil[1], Gaston R. Demarée[3], Andrea Kiss[4,5], Petr Dobrovolný[1], Kateřina Chromá[2],
Miroslav Trnka[2,6], Lukáš Dolák[1], Ladislava Řezníčková[1,2], Pavel Zahradníček[2,7], Danuta
Limanowka[8], Sylvie Jourdain[9]

[1]Institute of Geography, Masaryk University, Brno, Czech Republic
[2]Global Change Research Institute, Czech Academy of Sciences, Brno, Czech Republic
[3]Royal Meteorological Institute of Belgium, Brussels, Belgium
[4]Institute for Hydraulic Engineering and Water Resources Management, Vienna University of
Technology, Vienna, Austria
[5]Department of Historical Auxiliary Sciences, Institute of History, University of Szeged, Hungary
[6]Department of Agrosystems and Bioclimatology, Mendel University in Brno, Brno, Czech
Republic
[7]Czech Hydrometeorological Institute, Brno, Czech Republic
[8]Rydla 17, Kraków, Poland
[9]Météo-France, Direction de la Climatologie et des Services Climatiques, France

*Correspondence to*: Rudolf Brázdil (brazdil@sci.muni.cz)

**Abstract.** Extreme droughts are weather phenomena of considerable importance, involving
significant environmental and societal impacts. While those that have occurred in the comparatively
recent period of instrumental measurement are identified and dated on the basis of systematic,
machine-standardised meteorological and hydrological observations, droughts that took place in the
pre-instrumental period are usually described only through the medium of documentary evidence.
The extreme drought of 1842 in Europe presents a case in which information from documentary
data can be combined with systematic instrumental observations. Seasonal, gridded European
precipitation totals are used herein to describe general DJF, MAM and JJA precipitation patterns.
Annual variations in monthly temperatures and precipitation at individual stations are expressed
with respect to a 1961–1990 reference period, supplemented by calculation of selected drought
indices (SPI-1, SPEI-1 and Z-index). The mean circulation patterns during the driest months are
elucidated by means of SLP maps, NAO and CEZI indices. Generally drier patterns in 1842
prevailed in January–February and at various intensities between April and August. The driest
patterns in 1842 occurred in a broad zonal belt extending from France to eastern central Europe. A
range of documentary data is used to describe the peculiarities of agricultural, hydrological and
socio-economic droughts, with particular attention to environmental and societal impacts and
human responses to them. Although overall grain yields were not very strongly influenced, a
particularly bad hay harvest, no aftermath (hay from a second cut), and low potato yields led to
severe problems, especially for those who raised cattle. Finally, the 1842 drought is discussed in
terms of long-term drought variability, European tree-ring-based scPDSI reconstruction, and the
broader context of societal impacts.

## 1 Introduction

Dry events, generally caused by reductions in precipitation totals compared to normal climatic
conditions in a given area (meteorological drought), do not usually have such immediate and
dramatic consequences (e.g. immediate loss of human lives, material damage) as might result from
other hydrometeorological extremes – torrential rain, hailstorms, windstorms, floods, etc. The
impacts of droughts appear over time, with some delay in the case of meteorological drought and
progressively in agriculture (agricultural drought), water resources (hydrological and underground
water drought), and society (socio-economic drought) (Heim, 2002; Mishra and Singh, 2010;





Wilhite and Pulwarty, 2018). The anthropogenic activities in origins of certain droughts should be not omitted (Van Loon et al., 2016a, 2016b).

  Particular attention to individual extraordinary droughts events, based on the use of instrumental meteorological and hydrological data, is reflected in the considerable numbers of papers published in recent years (e.g. Fink et al., 2004; Shmakin et al., 2013; Hoerling et al., 2014; Spinoni et al., 2015; Zahradníček et al., 2015; Kogan and Guo, 2016; Laaha et al., 2017; Uhe et al., 2018). Contributions combining instrumental measurements with documentary data in the analysis of such events are less frequent (e.g. Dodds et al., 2009; Brázdil et al., 2016). Studies based exclusively upon documentary data tend to be those analysing severe drought events in the pre-instrumental period, for example, in Europe (Wetter et al., 2014; Kiss and Nikolić, 2015; Roggenkamp and Herget, 2015; Kiss, 2017, 2019; Pfister, 2018; Camenisch et al., 2019) or in other parts of the world (e.g. Hao et al., 2010; Zhang and Liang, 2010). Droughts in the pre-instrumental period may also be based on other types of proxies, particularly tree-rings (see mainly PAGES Hydro2k Consortium, 2017 for an overview).

  However, there are only few historical droughts that might lead to such an immediate response as that elicited by the 1842 drought. In particular, as early as 1 October 1842, Moritz Beyer (1807–1854), Professor of Agriculture at the Collegium Carolinum in Braunschweig, Germany, wrote of it in the preface to *Futternoth- und Hülfsbuch* ("The Book of Help in Forage Poverty"), motivated by the way in which the recent extreme drought had resulted in a critical lack of forage for livestock. He gave an overview of corresponding effects in many central European countries, and beyond, as well as recommendations as to how to mitigate negative effects (Beyer, 1843). The annual report of the I. R. Patriotic-Economic Society in Bohemia, in an evaluation of the agricultural conditions for 1842 mentioned that "*the adverse consequences of extraordinary atmospheric patterns in individual parts of agriculture will remain appreciable* [for a] *long* [time]" (Neue Schriften, 1845, p. 216). The extraordinary drought of 1842 was reported in several papers: for example, for south-eastern Poland and north-western Ukraine (former Galicia; Szewczuk, 1939), Polish Silesia (Inglot, 1968), the Czech Lands (Munzar, 2004; Brázdil and Trnka, 2015; Brázdil et al., 2019a), and for the north-eastern part of the Greater Alpine Region (van der Schrier et al., 2007).

  Developing upon the above, the current paper presents a comprehensive spatiotemporal analysis of the 1842 drought on a European scale (apart from the Mediterranean), based on the use of documentary evidence and systematic meteorological and hydrological observations that facilitate description of various aspects of this event. Section 2 characterises the basic sources of documentary data, instrumental measurements and other types of data used. The methodology of the paper is described in Section 3. The results in Section 4 concentrate on the spatiotemporal aspects of the 1842 drought, the synoptic and circulation patterns leading to it, and human impacts and responses. The longer-term context of this drought and its reflection in tree-rings and phenophases, as well as the broader context of its impacts are presented in Section 5. The final section contains some concluding remarks.

## 2 Data

### 2.1 Documentary data

Information concerning the 1842 drought may be derived from a range of documentary sources available from a number of European countries, as below:

### (i) narrative sources (chronicles, "books of memory", diaries)

Narrative sources usually report the occurrence or lack of precipitation and describe drought impacts for individual events on a broad scale. For example, the year 1842 appears in a chronicle kept in the bible of the Pitas family from Bohdašín (Bohemia) (Robek, 1976, p. 48): "*It was such a dry year that from the month of April until the month of December only three light rains occurred, such that roots did not get wet. By the Will of God, the rye still recovered, but spring crops were so sparse that people had to pluck them in many places. The worst* [situation] *was for the cattle, because neither fodder nor clover grew on grazing land, and the grasslands were so dried out that*



*they turned red, as if burnt by fire. Wells and brooks dried out and* [matters were] *so bad for milling that people had to carry* [grain] *several miles to mills with higher water.*" In a diary kept by Doeke Wijgers Hellema (1766–1856), a teacher in Wirdum (Friesland, The Netherlands), one may read: "*22 August 1842, although not as dry and warm as before, is still just as dry, even today. Although*
*the sky was somewhat overcast last night, there is now almost clear air once more in the morning. The earth thirsts for water. Some lands here, perhaps more so elsewhere, are as dry as mid-winter. In addition, most ditches are empty, and farmers can hardly provide their animals with drinking water*." (Archival source AS2, p. 86).

**(ii) weather diaries and related weather observations**
Certain people of education – county or state officials such as medical officers, town or district physicians, doctors and priests wrote monthly weather reports, based on their weather diaries (sometimes with temperature and/or pressure measurements), as parts of the reports of illnesses they submitted to administrative bodies (Réthly, 1999). These departments, responsible for health issues, often required such information in order to prevent, monitor and limit the proliferation of diseases
(e.g. Berde, 1847). For example, the weather diary kept by Mihály Király, a subnotary of the Evangelical church in Agârbiciu, central Transylvania (Romania), contains the following direct comments on droughts (Réthly, 1999, pp. 781–784): [24 April]: "[…] *silent, cloudy weather in the morning, and then strong, dry, cold wind. The vegetation is hampered and animals suffer as there is neither hay nor grass to feed them*."; [13–15 August]: "[…] *continuous great heat, mostly windy*
*days, very cool nights and mornings. Horrifying drought and huge* [quantities of] *dust; vegetation and harvest hampered in all senses*"; [15 September]: "[…] *enormous thunderstorm at dawn, which eased the sweltering ground in this extraordinary three-month drought, which* [in this area] *stands almost without precedent, and gave new life to everything.*" Similar types of report were also required from (hydro) engineers with respect to the weather and the daily water levels of
particularly problematic rivers (see e.g. the Tisza – AS1).

**(iii) newspapers**
Newspapers contain information that covers droughts, especially when impacts on the environment and society are especially severe. In particular, they report on the dry state of the soil and the lack of available water, on the harvests of various crops, the prices of agricultural products and foodstuffs,
the outbreaks of fires, etc. For example, on 7 June 1842 the French *Journal des Débats politique et littéraire* reported (p. 2) on the danger of a bad harvest due to heavy drought: "*We read in the Précurseur de l'Ouest* [at Angers] *for 4 June: 'We have received sad information on the state of the countryside and the future of the crops from all points of the department of Maine-et-Loire. Hay, wheat, grain, and plants of all kinds have suffered considerably from the drought. The meadows*
*will give little or no grass; the wheat is already yellowing before reaching its ordinary growth; oats are puny and short; the flax has aborted and will give only seed. Hemp, newly sown, will be in great danger.*" For Tápiószentmiklós in northern Hungary, *Nemzeti Újság* (10 Sep. 1842, p. 291) described the situation in August 1842: "*Complaints about the harvest and drought of the current year. The winter crops gave 2.5 seeds on average, the oats 2, and the barley 3–4 seeds. We will*
*perhaps have potatoes: there might have been ample fruiting, but* [the foliage became] *burnt. The pasture is narrow* [i.e. bad]*. There are only two wells in this huge area, but even there* [i.e. near them]*, there is only little water. There will probably be wine.*" Reports from places other than the area or country of publication very often appear in newspapers. For example, a report relating to Wrocław (Poland) appears on 23 August in the Austrian *Wiener Zeitung* (28 Aug. 1842, p. 1770):
"*Since 1811 no year has approached* [this] *one in terms of dryness* [1842], [a fact that] *is at least valid for our province. Although the year 1834 was almost drier than the current* [year] *for the eastern countries of Europe, we did not suffer so much from rain deficiency, and therefore we had no such paucity of fodder as in this year. Considerable shortage holds sway over many economies already, and it will intensify steadily until next spring* [1843].*"

**(iv) epigraphic evidence**
Low water levels on rivers following drought episodes may be recorded as marks for the corresponding year on "hunger stones" (also known as "dearth stones"), names that reflect failures





of harvests and consequent increases in food prices, together with hunger, deprivation and poverty. They are well known, for example, in the Czech Lands (Brázdil et al., 2013), Germany, Hungary (Palotay et al., 2012) and Switzerland (Pfister et al., 2006), but signs of the 1842 drought appear only on the River Elbe at Děčín in Bohemia (Fig. 1) and Magdeburg and Dresden in Germany (Elleder, 2016).

**(v) society reports**

A number of scientific and economic societies were interested in weather patterns with respect to their effects on agriculture, horticulture and forestry, even organising their own meteorological observations. For example, the *Société nationale d'horticulture de Paris* in France published the results of meteorological observations made by M. Jacques, horticulturist to the Duc d'Orléans, referring to the period between 1832 and 1850 in Neuilly-sur-Seine Château de Villiers (Société nationale d'horticulture de Paris, 1842). Two remarks made in the context of his instrumental measurements reflect the drought of 1842: "[…] *drought is extreme this year; vegetables are scarce and expensive. The tree nurseries are affected by this terrible drought and the trees are growing poorly.*" (July 1842, p. 137); "*Finally, after 5 months of terrible drought, rain that fell in early September did a world of good* […]" (September 1842, p. 223). In Bohemia, the I. R. Patriotic-Economic Society, as well as organising and publishing results of meteorological and phenological observations from its own network of stations (Brázdil et al., 2011; Bělínová and Brázdil, 2012), reported annually on the general state of agriculture and forestry. For 1842, bad yields of agricultural crops in Bohemia and their consequences were evaluated as follows (Neue Schriften, 1845, pp. 218–219): "*The harvest started earlier than usual everywhere; yields were, for the greater part, particularly of rye, below average; for barley and oats bad; and only for wheat a little better. Potatoes gave a bad harvest in almost the entire country* […] *nearly everywhere the harvest of hay* [in 1842] *was less than that of 1841 by a third or a half. The aftermath* [hay from a second cut] *failed totally nearly everywhere* […] *Reduction of cattle numbers by a third was a necessary consequence of lack of fodder in some places, in which case cattle were taken to slaughter at very low prices due to sudden saturation of the market. This harm* […] *will be experienced strongly in high prices of meat and dairy products, as well as in reduction of manure for soil fertility over time.*"

**(vi) professional journals**

Weather conditions and their detectable consequences, largely in the form of monthly overviews, were regularly published by professionals, mainly in agricultural and medical journals, often linked to specialist societies within a given country; examples include *Magyar Gazda* (Hungarian Farmer) and *Orvosi Tár* (The Medical Collection) in Hungary. These regular reports related to various parts of the country. For example, the weather conditions in July and August 1842, with certain health consequences, were described by the local medical officer in Uzhorod (Ukraine) in what was then Ung County (*Orvosi Tár*, vol. 3/3, pp. 152–153): July: "[…] *There was great drought from the beginning of the month until the 22nd, and only in the last week* [of the month] *was there some rain that revived our vegetation.*" August: "[…] *In this month, except for the 1st, 7th and 27th, the weather was very clear, dry and pleasant; light winds in the morning and evening alleviated the great heat. / Medical conditions: In the hot and dry days of this month we continually observed* [a rising number of] *painful cases of urinary sand which, combined with incessant and prevailing intermittent fever (malaria), had even more desperate consequences* […] *Among children, there was an epidemic of bronchitis* […]"

**(vii) professional papers**

In the wake of certain outstanding extremes (particularly floods or storms), several kinds of professional papers related to such events appear. An example of a publication by Beyer (1843), related to the 1842 drought has already been pointed out in Sect. 1.

**2.2 Instrumental series**

National meteorological services were established in the second half of the 19th century (e.g. in 1851 in the Austrian empire – Hammerl et al., 2001). Thus the year 1842 belongs rather to the



period of early instrumental meteorological observations, despite the fact that a number of networks of meteorological stations were already in existence (see e.g. Bělínová and Brázdil, 2012, for Bohemia). Although various databases of meteorological measurements have been created for Europe (e.g. ECAD – see https://www.ecad.eu//dailydata/predefinedseries.php; last access: 19 April

2019), systematic homogenisation of these observations is a relatively recent phenomenon (http://surfobs.climate.copernicus.eu/surfobs.php; last access: 19 April 2019). The HISTALP database (Auer et al., 2007), covering the Greater Alpine Region, is currently only one of a few publicly-available sources of homogenised monthly temperature and precipitation series holding data concerning the period before 1842 (http://www.zamg.ac.at/histalp/index.php; last access: 19

April 2019). Such larger-scale endeavours aside, certain individual researchers and institutions have also contributed to the homogenisation of long-term temperature and precipitation series in some European countries.

There exist a number of long-term hydrological series of water levels (from which discharges may be calculated later), and they include data for 1842. One example may be found in a

paper by Novotný (1963), who created series of mean monthly discharges for the River Vltava at Prague (Bohemia) for the 1825–1957 period. Further, daily water levels of the River Danube at Vienna (Austria), for example, were published in *Wiener Zeitung*, for two profiles. One of these was situated at the Great Danube Bridge (*Grosse Donaubrücke*, demolished as part of Danube regulation works in 1870–1875, which was close to the current *Floridsdorfer Brücke* – Rohr, 2019),

where recording started on 1 April 1826. The other was on Danube channel (*Franzensbrücke*) since 1 January 1815. Both used units expressed in Vienna feet and inches (1 Vienna foot = 12 Vienna inches = 31.608 cm, 1 Vienna inch = 2.634 cm). Water levels were also measured and published in *Jelencor* for the Danube at Budapest (Hungary), but until July 1842 measurements were only taken once or twice a week, and only occasionally thereafter. Despite the availability of figures for water

levels on the other European rivers, and their particular use for the study of flood events, it is difficult to find daily water levels for 1842 for further rivers that are, at the very least, comparable with their adjacent years.

**2.3 Other data sources**

A spatial overview of seasonal precipitation patterns throughout Europe may be obtained from reconstructed totals for the 1500–2000 period calculated by Pauling et al. (2006). This reconstruction combines gridded values (0.5º x 0.5º) of the European lands (30ºW–40ºE, 30–71ºN) for the years 1500–1900 with the results of a gridded reanalysis for 1901–2000. The reconstruction is based on long, instrumental precipitation series, documentary-based precipitation indices and

natural proxies sensitive to precipitation signal, such as tree-rings, ice cores, corals and speleothems.

Sea-level pressure (SLP) patterns in the Atlantic-European area were examined in terms of the database of gridded SLP created by Luterbacher et al. (2002) (https://www.ncdc.noaa.gov/paleo-search/study/6366: last access 19 April 2019). Atmospheric

circulation was examined in terms of the North Atlantic Oscillation Index (NAOI) in two versions and the Central European Zonal Index (CEZI). The NAOI series, after Jones et al. (1997), taken from the CRU website (https://crudata.uea.ac.uk/cru/data/nao/; last access: 19 April 2019), calculates on the basis of data from Gibraltar and the Reykjavik area of Iceland, while NAOI after Luterbacher et al. (1999) takes into account normalised differences between SLP means of four 5° x

5° (latitude/longitude) grid-points over the Azores and over Iceland. CEZI is calculated as the difference in the standardised mean SLP, averaged for the grid points 35°N/0°, 35°N/20°E, 40°N/0°, 40°N/20°E and 60°N/0°, 60°N/20°E, 65°N/0°, 65°N/20°E (Jacobeit et al., 2001).

Indicative fluctuations in the prices of agricultural products and goods were derived from series of prices for wheat, rye and barley over the 1655–1872 period collected for Prague (Czech

Lands) by Schebek (1873). Moreover, many newspapers have always reported on the market prices of various agricultural crops (particularly grain), as well as other commodities.



## 3 Methods

Documentary data extracted from the basic sources described in Sect. 2.1 were first critically evaluated with respect to the credibility of sources, their dating and the contents of the reports. Obsolete place-names were converted into their recent, more accessible form and the corresponding
country added. Documentary data, by nature of a qualitative character, were employed, in particular, to descriptions of the character of hydrological, agricultural and socio-economic droughts from the point of view of environmental and societal impacts, as well as human responses. Further to such general characterisation, descriptions of droughts were supplemented by numerous examples from other European countries. However, space and purpose have dictated that only a
small proportion of the huge amount of information collected could be included in this paper. Original narrative sources from the archives have been quoted only in the case that they have not been previously translated into English, edited or published.

Gridded totals over the lands of Europe by Pauling et al. (2006) were used to show spatial precipitation distribution in winter (DJF) 1841/1842, spring (MAM), summer (JJA) and composite
winter–summer (DJF–JJA) seasons for 1842. The corresponding totals were expressed as deviations from the 1961–1990 reference period seasonal means on the one hand and as percentages of these reference means on the other (Fig. 2). Daily mean temperatures, daily precipitation totals, and their cumulative values for the Prague-Klementinum station (Fig. 3) served to illustrate weather changes during 1842. A range of long-term monthly temperature and precipitation series put the year in
question in perspective in terms of annual variation. To examine temperature, monthly variations during 1842 were presented as deviations from the 1961–1990 reference period, while the monthly precipitation totals for 1842 were expressed as percentages of the corresponding reference (Fig. 4). These series were then worked up to calculate monthly drought indices: (i) Standardised Precipitation Index for one month – SPI-1 (McKee et al., 1993); (ii) Standardised Precipitation
Evapotranspiration Index for one month – SPEI-1 (Vicente-Serrano et al., 2010); and (iii) Z-index (Palmer, 1965) (Fig. 5).

In order to investigate the synoptic patterns current in the driest months of 1842 (January, February, April–August), monthly SLP maps for the Atlantic-European area were created and also expressed as deviations from the SLP means in 1961–1990 (Fig. 6). These maps were supplemented
by the annual variations in mean monthly NAOI and CEZI, in order to investigate any possible further circulation involvement with droughts (Fig. 7).

In the light of their general similarities, the features common to the general impacts of hydrological, agricultural and socio-economic droughts in Europe are always mentioned. This general information is supplemented by many examples from documentary evidence reporting the
individual features of droughts, their consequences and human responses to them. Moreover, measurements of water levels or calculated discharges expressing their annual variations have been used for further characterisation of hydrological drought (Fig. 8).

## 4 Results
### 4.1 Spatiotemporal variability of the weather in 1842
### 4.1.1 Precipitation patterns

The spatial DJF, MAM and JJA precipitation patterns for the driest parts of the 1842 year over the European lands, based on gridded precipitation totals by Pauling et al. (2006), appear in Fig. 2. Totals for 1842 are expressed both as deviations from the reference 1961–1990 seasonal means and
as percentages of means derived from this reference period. Dry patterns in DJF appear especially from eastern France to the western part of central Europe and then in a broad area around the Black Sea (Fig. 2a). DJF precipitation shows decreases of more than 20% compared to the reference means, creating a broad belt stretching from eastern France to eastern Europe (Fig. 2b). Particularly dry in MAM was a band of territory stretching over western Europe from southern Scandinavia to
northern Spain, including a large part of the British Isles (Fig. 2c), with totals more than 20% below the reference means. Other areas with relatively marked MAM precipitation decreases extended from the Black Sea to Poland, further in the Balkan region close to the Adriatic Sea, and in north-



eastern Europe (Fig. 2d). Lower precipitation in JJA extended from the north of the British Isles to
southern Scandinavia and notably from France, then over central to eastern Europe (Fig. 2e). Areas
from the North Sea running south-east, with slightly lower totals in a belt from north-western
Poland to southern Germany were the driest in relative terms (more than 20% below the JJA
reference means) (Fig. 2f). The distribution of JJA precipitation influenced the picture for
composite DJF–JJA patterns (Fig. 2g), in which a dry belt from south-east Scandinavia to north-east
France predominated in relative expression (Fig. 2h). However, generally lower DJF–JJA
precipitation totals were experienced in a broad band extending from the British Isles and France to
eastern Europe (Fig. 2h). SON precipitation (not shown) demonstrated somewhat wetter patterns to
the south of *c*. 52–53ºN and drier patterns to the north of this latitude.

**4.1.2 Annual variations in temperature, precipitation and drought indices**
Fluctuations in daily mean temperatures and daily precipitation totals during 1842 at the Prague-
Klementinum station appear in Fig. 3. Compared with the 1961–1990 reference period,
temperatures remained below their corresponding means for the greater part of the year, particularly
in January and February, the first part of April and at the turn of October/November. A very
consistent cold period persisted from the second half of September to early December. Clearly
warmer patterns compared to the 30-year mean occurred only in August (Fig. 3a). The highest daily
precipitation total achieved only 13.8 mm, on 8 June. The deficit in precipitation totals,
continuously growing from mid-April to the end of 1842, finally reached 215 mm compared to the
30-year mean of 470 mm (Fig. 3b). Helping to complete the picture, the annual frequency of
precipitation days with a total of ≥0.1 mm in 1842 achieved 126 days compared with the mean of
151.7 days in 1961–1990. In terms of frequencies of days with totals of 0.1–1.0, 1.1–5.0, 5.1–10.0
and ≥10.1 mm, the differences in precipitation days between 1842 and the reference period were at
their highest for the last-mentioned interval (2 days against 10.4 days). During the driest period,
from April to August 1842, only 30 precipitation days were recorded compared with 67.7 days in
the 1961–1990 period. August was outstandingly dry, with only 2 precipitation days compared to
the mean of 13 days for this month.
      Documentary sources usually report extended periods without rain, or when the first rain
falls after a dry period. For example, on Greenock and Glasgow in Scotland, renowned even in that
proverbially moist land for their damp and rainy nature, not a single raindrop fell in the whole of
April (*Wiener Zeitung*, 28 May 1842). In central Transylvania the *Siebenburger Bote* (26 May
1842) reported a drought that lasted until 22–25 May, when torrential rain led to a flash flood, but in
other areas drought continued without the usual late-May rain (*Erdélyi Híradó*, 21 June 1842). To
emphasise the extremity of such weather, reports often add information about high temperatures.
For example, a story dated 1 September from Trieste in Italy reported that since mid-June there had
been virtually no rainfall in the vicinity and temperatures were almost constantly between 22 and 25
degrees Réaumur (°R), i.e. 27.5°C–31.2°C (*Journal de Bruxelles*, 15 Sep. 1842). In Troitsk, Russia,
not a drop of rain fell from the end of June onwards and on 27 July temperatures rose to 28°R to
32°R (i.e. 35°C–40°C). This heat, together with dry westerly winds, desiccated the soil to a depth of
150 cm (*Le Messager de Gand*, 19 Oct. 1842). Although such daily entries have been reported for
many European places, they lack sufficient systematic character to cite.
      To examine the annual structure of weather in 1842, certain European meteorological
stations or regions were selected to describe temperature and precipitation on the basis of monthly
45   values (Fig. 4). Despite existing regional differences, cooler and drier patterns prevailed in January–
February. After above-normal March precipitation, dry patterns prevailed at various degrees of
intensity from April to August. August was also the month with the highest positive temperature
deviation.
      Stations and regions appearing in Fig. 4 (except for the Swiss Plateau, which was replaced
50   by Vienna because data was absent) were used to calculate drought indices expressing the actual
drought, namely SPI-1, SPEI-1 and Z-index (Fig. 5). Indices are expressed in real values and their
negative figures reflect drier patterns. While SPI-1, expressing variations of precipitation totals,



indicates a great lack of precipitation (see e.g. the lowest values for April in Stockholm and for August in Warsaw), SPEI-1 combines the effects of precipitation with temperatures, from which evapotranspiration is estimated. However, annual variations in both SPI-1 and SPEI-1 indices are very similar. The third indicator, the Z-index, used for calculation of Palmer Drought Severity

Index (PDSI) and representing soil moisture patterns, tends to show a consistent dry period, as seen for JJA in Trieste and AMJJA in Paris, Karlsruhe, and Vienna, as well as the Czech Lands, for which generally increasing intensity from April to August is apparent. In Edinburgh, the drought persisted unremittingly from April to December.

**4.2 Synoptic patterns in 1842**
SLP patterns in the Atlantic-European area for January, February and all months between April and August 1842, and their deviations with respect to the corresponding 1961–1990 reference period appear in Fig. 6. In January, eastern and central Europe and part of Scandinavia were under the influence of high pressure from the east, becoming far weaker towards south-western Europe

(Fig. 6a). In February, a considerable portion of Europe was under a ridge of high pressure, with the anticyclone located to the north and north-west of the Black Sea, and a high pressure system south-west of the Iberian Peninsula (Fig. 6c). In April, an extensive high pressure system centred over the northern British Isles influenced weather patterns over the greater part of Europe (Fig. 6e). In May, a pressure col extended over much of western Europe, running between a high pressure area south-

west of the Iberian Peninsula and another high over eastern Europe (Fig. 6g). In June, most of Europe lay within a belt of high pressure extending from the Azorean High, located far south-west of the Iberian Peninsula; a distinct low appeared in northern and north-eastern Europe (Fig. 6i). A similar distribution of SLP patterns held for the following month of July (Fig. 6k). In August, a large part of Europe lay within a belt of high pressure running from south-west to north-east, with

the Azorean High west of the Iberian Peninsula and an isolated anticyclone over the Baltic Sea (Fig. 6m). All over Europe, in all the months analysed excepting July (Fig. 6l), an area of considerable pressure increases compared to the 1961–1990 means was evident in one area or another (Fig. 6b,d,f,h,j,n).

Other available circulation indices were also used to describe relevant circulation patterns in

Europe in 1842 (Fig. 7). According to Jones et al. (1997), a clearly-expressed positive NAOI mode prevailed in January–March and negative modes in June–July and September–November. According to Luterbacher et al. (1999), there were remarkably well-expressed positive NAOI modes in February–March, May and August, while negative modes were clearly pronounced in April and also in the SON months. Analysing CEZI (Jacobeit et al., 2001), we found a positive

mode predominant in March and a negative in September. Because positives modes of all three indices reflect rather western airflow from the Atlantic Ocean into the studied part of Europe, Fig. 7 demonstrates complicated relationship of precipitation/drought patterns to circulation indices.

**4.3 Impacts and human responses to the 1842 drought**
**4.3.1 Hydrological drought**
The weather patterns of 1842 were reflected in low water levels and discharges on several rivers. For example, the daily water levels of the River Danube at Vienna, published regularly in *Wiener Zeitung*, were recorded for the main river at the Great Danube Bridge as well as on the river channel (see Sect. 2.2). Since both profiles express identical changes, Fig. 8 shows fluctuations in daily

water levels at the bridge profile in 1842 in comparison with the corresponding levels from 1841 and 1843. After very low water levels in January–February 1842, related to the channels being frozen over, water levels were below those of the two adjacent years in June–July and then from mid-August to around the beginning of November, with a very strongly expressed declining trend. A further considerable decrease appeared around mid-December. The mean annual discharge of

3342 $m^3 s^{-1}$ for the year 1842 is the eleventh lowest value in the long-term annual series of Danube discharges at Vienna in the 1828–2011 period. Tendencies similar to those observed in Vienna are recorded for the Danube at Budapest: from not particularly high water levels in May and June, there



was a steady decrease, with a particular drop in early July; water levels then continued to remain somewhat low, although with minimal decrease. Very low water levels appeared in December (*Nemzeti Újság*, 21 Dec. 1842).

On the River Vltava in Prague, very low discharges were recorded during a dry January–February period (48 and 46 m$^3$ s$^{-1}$ respectively). After a steep increase to the annual maximum in March (255 m$^3$ s$^{-1}$), they began to decrease (the most dramatic drop – by 119 m$^3$ s$^{-1}$ – occurred between April and May) to their lowest values in July–September, with the annual minimum in August (35 m$^3$ s$^{-1}$). A slight increase followed (Novotný, 1963). Very low water stages were also recorded on the River Elbe, with epigraphic testimony appearing as a mark for 1842 on the hunger

stone at Děčín-Podmokly in Bohemia (see Fig. 1). The emergence of a similar hunger stone on the German part of the Elbe in Dresden and "*the near disappearance*" of this river around Pirna was also reported in *Lemberger Zeitung* (9 Sep. 1842). The waters of the rivers in south-eastern Poland ran at unusually low levels and some rivers even dried out (Szewczuk, 1939). The exceedingly low water level recorded for the River Tisza in Szeged (Hungary) in 1842 later became the zero point

for water level measurements at that point (Botár and Károlyi, 1971). Records of daily water levels there suggested a steady decrease from April, with the lowest values in August and September (AS1). On the River Seine at Paris, the water level dropped for 52 days below the zero point on the Pont Royal scale, established already in 1719 (Fuster, 1845).

         Low water levels in the Neusiedlersee/Fertő-tó, today located on the Austrian-Hungarian

border, were also noted for 1842 (Élő, 1937); the planning and ground investigations for the drainage of the South Balaton wetlands, which usually took place at times of low water level, were carried out in this year (Szaplonczay, 1914).

         A very rich body of documentary sources report a general lack of water in 1842, when many wells and springs dried out. Brooks and small watercourses ran dry. In Älvkarleby (near Gävle,

Sweden), one arm of a waterfall dried out completely and the main arm carried so little water that it was possible to cross it on foot (*Wiener Zeitung*, 21 Oct. 1842). Absence of water on even large rivers led to interference with commercial ship transport for several months. In France, shipping on the River Seine at Paris was disrupted for around four months (Fuster, 1845). A newspaper story dated 29 June from Regensburg (Germany) reports that the Bayern-Württemberg Steam Navigation

Company cancelled a cruise on the River Danube (*Linzer Zeitung*, 6 July 1842). A similar situation pertained to almost all the usually-navigable German rivers. From Dresden, reports indicate that transport on the River Elbe was suspended on 11 July. Cessation of steamship traffic is also reported for the River Main at Frankfurt am Main on 5 July, and a similar situation was mentioned for the River Mosel (*Allgemeine Preußische Staats-Zeitung*, 9 July and 14 July 1842). The ferry

between Rotterdam and Venlo (The Netherlands) had to be suspended due to persistent low water levels on the river Maas/Meuse which "*will became non-navigable if the current unprecedented drought does not cease soon*" (*Algemeen Handelsblad*, 2 Sep. 1842, p. 1). As the drought went on, the levels of the River Weser in north-western Germany fell into a state not seen for 200 years. An enormous quantity of dead fish led to a foul odour of putrefaction, and their remains had to be

removed by the cart-load (*L'Indépendance Belge*, 2 Sep. 1842). It was possible to wade across even such large rivers.

         Problems arising out of scarcity of water were especially acute in towns and cities. When a general and urgent lack of drinking water started to occur in Middelburg, the capital of the Zeeland peninsula in The Netherlands, the town authorities found themselves obliged to send a ship to

"mainland" Holland on 8 June in order to provide it (*Nederlandsche Staatscourant*, 11 June 1842). According to a report from the city of Goes on the island of Zuid-Beveland (The Netherlands) dated 13 June, good drinking water was so scarce that many people were using bad water, and adverse health consequences were feared (*Vlissingsche Courant*, 17 June 1842). A report that appeared on 31 August indicated that one had to travel 7 miles (*c.* 51 km) from Hamburg (Germany) to

Glückstad, higher up the Elbe, or 4 miles (*c.* 29 km) to Stade for drinking water because local springs were completely dry and the lower Elbe water was too brackish to be used for drinking (*Algemeen Handelsblad*, 5 Sep. 1842). Even the Alps had water problems, as reported, for example





in Stuttgart (Germany) on 20 April (*Allgemeine Preußische Staats-Zeitung*, 27 Aug. 1842, p. 1015): "*The worst* [of it] *is for the inhabitants of the Alps, where a lack of water has achieved the highest degree, and true caravans with water tanks travel several hours* [down to] *to valleys to ensure necessary water.*" In Trieste (Italy), based on a report dated 1 September, more than 300 public
water-points ("fountains") were dry. Serving maids were forced to wait for hours at the few fountains that still had water, before they could fill their buckets (*Journal de Bruxelles*, 15 Sep. 1842).

Lack of water was fatal to the operation of water mills and many were simply forced out of operation. A concise description of the problems facing milling was given by the chronicler Václav
Krolmus, for Bohemia (Krolmus, 1845, pp. 127–128): "*Two-thirds of* [water] *mills were standing* [idle in the] *drought. It went badly for milling; lone* [i.e. isolated] *millers sought mills on larger rivers where they could grind grain for their own houses. People had to travel 6–7 hours to distant rivers such as the Elbe, Jizera, Vltava, Ohře and Mže* [now Berounka]. *Further,* [even] *on large rivers, millers were forced to allow time for sufficient water to build up and people had to wait*
*several weeks* [for milling]*; all this was all the more* [valid] *on brooks that had only weak and tiny active springs. At that time millers in some places bought water from the large ponds owned by the nobility and collected enough to make milling for a few hours possible. Millers milled for one another.*"

**4.3.2 Agricultural drought**
When addressing plant growth, documentary sources tend to give preferential attention to rain events and their abundance. For example, Anton Pejšek from Zlončice (Bohemia) mentioned the drought of 1842 and problems for crops in these terms in several cases: the first rain of spring on 4 June wetted the soil to a depth of one (Vienna) inch (*c.* 2.6 cm). As late as 2 August, the soil was
wetted to the extent of "*a tiny little furrow*"; on 17 September this grew to three-quarters of a furrow and by 2 October had extended to the whole furrow, so that autumn sowing went well (Robek, 1958a). Lack of rain was the anticipated forerunner of a bad harvest, as documented by the French *La Patrie* (16 June 1842, p. 3): "*Ordinarily, says the Patriote de la Meurthe et des Vosges* [at Nancy]*, the month of June is rainy in our countryside; this is not the case this year: the pleasant*
*weather is a real calamity. For two months, hardly a drop of water has fallen on our plains and in our valleys. Hay harvesting will be less productive than a year ago. Wheat suffers from this pitiless drought; the vine alone maintains a beautiful appearance. May we soon see some clouds in this sky, the* [constant] *serenity* [of which] *is desperate.*" A similar situation was reported from north-eastern England (*The Morning Chronicle*, 23 June 1842, p. 6): "[...] *the crops are light, and if we are not*
*favoured with moisture from the clouds in a short time, the whole of the hay crops will prove extremely bad, pastures are scalding, and feed is much wanted – the potatoes and turnips have an unhealthy appearance, the former past recovery – the spring crops may be a fair average if they receive sufficient moisture by the end of the week.*"

The impacts of droughts in agriculture are particularly reflected in crop yields. For example,
documentary sources from the Czech Lands reported the following for 1842: weaker yields of barley and oats (the ears were sometimes so sparse and small that they had to be been harvested manually, without tools); wheat and rye yields were generally average depending on location; lack of straw, which was already short in the stem; failure of potatoes ("[potatoes] *were the size of walnuts and almost inedible,* [more like] *overgrown peas*" – Robek, 1978, p. 49); a meagre harvest
of vegetables (peas, lentils, cabbages), flax, beet, vetch and hops; unripe fruits fell from their trees early; it was impossible to plough, and stubble fields remained unturned after the harvest. There was a generally small yield of hay and no aftermath (i.e. the second hay-growth). A report dated 1 September from Trieste (Italy) noted that all the plants had died and trees were even losing their leaves, as if it were already autumn (*Journal de Bruxelles*, 15 Sep. 1842). Further east, in the Košice
area of Slovakia, cabbages and potatoes were in a poor state in the first days of September, due to extended heat and outstanding drought (*Nemzeti Újság*, 10 Sep. 1842); problems with the late



development and a bad harvest of potatoes due to drought were also mentioned in Verőce and Miskolc in Hungary (*Nemzeti Újság*, 13 and 26 Nov. 1842).

However, regional difference in yields could be quite marked. Records from Hungary (as then politically defined) indicate that the harvest of both winter and spring crops was weak, except
in the most significant grain production areas in the southern and eastern parts of the Great Hungarian Plain, particularly the Bačka (north Serbia) and the Banat (north-eastern Serbia and south-western Romania) regions (*Jelenkor*, 3 Sep. 1842). While rather bad cereal and hay harvests were reported in the northern part of the plain (e.g. *Nemzeti Újság*, 10 Sep. 1842), reports from around Kecskemét in its central part indicate that maize, barley and oat harvests were not good;
chronicles kept by Franciscan monks suggest a lack of these crops, while remaining silent about other harvests in 1842 (Szabó, 1992). In the Bratislava area of Slovakia in late July, when harvesters' wages were fixed, the spring crops were particularly bad, while the winter crops showed little promise over large areas (*Jelenkor*, 30 July 1842). Moreover, in Sáros County (north-eastern Slovakia), hail destroyed the harvests of more than 40 settlements (*Jelenkor*, 27 Aug. 1842).
Yields of grain also suffered from a population explosion of mice and hamsters. Some villages in the Mainz region (Germany) were driven to pay bounty on them. Similar overpopulation problems were reported for other regions of Germany, such as the Rhine region, Saxony, Thuringia and Lower Franconia (Beyer, 1843), but damage done by an overabundance of rodents was also reported in Hungarian Szekszárd (*Nemzeti Újság*, 19 Oct 1842). In Auchterarder (Scotland),
farmers lived in fear of "*myriads of small white worms*" which were capable of damaging all kinds of vegetables in time of drought (*The Morning Chronicle,* 23 June 1842, p. 6). Another interesting manifestation of insect overpopulation was reported from the Netherlands, where farmers in some places were unable to plough their fields in the face of a drought-facilitated multitude of wasps' nests (*Nederlandsche Staatscourant*, 18 Aug. 1842). In Neuchâtel, Switzerland, warnings appeared
in September of the potential for overpopulation of wasps (due to the warm, dry summer) with respect to the damage that they might do to grapes (*Wiener Zeitung*, 20 Sep. 1842).

In contrast to their negative influence on many other agricultural crops, high temperatures with drier patterns may prove positive to the quality of wine. Reports from Alsace (France) indicate that the grapes had to be picked and the soil bored in the dry times that which lasted from mid-
January to August. The cultivators had to use wedge-shaped hoes to split and pulverize soil so hard that it often damaged the tools. Because the vines themselves and other crops suffered heavily from the drought, the harvest was mediocre, or worse, in quantity; however, the wine was of excellent quality (Liblin, 1872). Beyond France, wine of excellent quality was also reported from other European countries such as Bohemia (Katzerowsky, 1895) and Moravia (Drössler, 1933). Generally
ample harvests of grapes were also reported in Hungary (e.g. in Veszprém, Budapest – *Nemzeti Újság*, 19 Oct. and 8 Nov. 1842), but in most areas this held true for quantity rather than quality. Hailstorms, cold winds and a rainy period from mid-September onwards were blamed for the medium quality in most places (e.g. Eger, Zemplén, Kőszeg, Verőce, Nagykálló).

**4.3.3 Socio-economic drought**

The problems with lack of water described in Sects. 4.3.1 and 4.3.2 also found reflections in socio-economic matters. That more frequent fires were reported for 1842 in several documentary sources is testimony to such negative effects. A catastrophic example may be found in the *Große Brand* ("Great Fire") of Hamburg. After several weeks of extreme drought, fire broke out on 5 May at 1
a.m. "*which took until the morning of 8 May to extinguish*" (for a detailed description, see Schleiden, 1843). Germany's most serious peacetime fire in modern history, it destroyed a third of the old town, with 72 streets, around 1100 houses and 102 granaries. There were 51 fatalities, 120 people were injured and 19,995 people were made homeless (Hamburger Feuerwehr-Historiker E. V., 2005). An account of a fire in Rzeszów (Poland) on the night 26/27 June notes: "*there was no*
*rain for many days and roofs were so dry that each spark could lead to a devastating fire*" (*Wiener Zeitung*, 10 July 1842, p. 1413). Lack of water complicated fire-fighting. For example, during a fire that broke out on 14 June in Zlončice in Bohemia, wells held so little water that people could only



stand by and watch the conflagration, with no means of extinguishing it (Robek, 1958a); a similar situation occurred during a fire on 23 August in Králova Lhota (Robek, 1978). In Austria, fire broke out on 17 July in Korneuburg; but what water remained in the wells after a long, dry and warm period was soon exhausted and three small lakes nearby were of little help (*Wiener Zeitung*, 21 July
5    1842).

The year 1842 also stands out in records of forest fires from many sources. For example, in the Branná domain (the Moravian part of Silesia), fire broke out on 17 August in the Liechtenstein Forest, burning for several days at the expense of over 2000 fathoms (*c*. 5680 m$^3$) of timber; it had still not been extinguished on 26 August (*Diekircher Wochenblatt*, 10 Sep. 1842). Blase (1845)
reported in great detail a forest fire that started on 31 August in "Saxon-Bohemian Switzerland" and went on to destroy a forest area of 327 "Acker" (*c*. 181 ha), of which 172 "Acker" were on the Saxon side and 155 on the Bohemian side (*c*. 95.2 ha and 85.8 ha respectively). A number of reports of August forest fires also appear for Norway (e.g. *Münchner Politische Zeitung*, 26 Aug. 1842; *Wiener Zeitung*, 15 Sep. 1842).
A general idea of the statistical scale of city and forest fires occurring in the generally driest period of April–September 1842 may be obtained if reports appearing only in *Wiener Zeitung* are considered. These indicate a total of 68 concrete cases of fire elsewhere in Europe nevertheless considered newsworthy by an Austrian newspaper (in Austria, Belgium, the Czech Lands, Germany, Hungary, France, Italy, The Netherlands, Norway, Poland, Romania, Russia, Sweden,
and England). Nine of the stories explicitly mention drought or lack of water. Moreover, in addition to such precisely localised fires, more general statements about further fires in the given region or country also appear (for example, a despatch from Paris, dated 28 September, makes wider mention of daily fires in the Seine-et-Marne Region – see *Wiener Zeitung*, 7 Oct. 1842).

The general lack of water (Sect. 4.3.1) had many further consequences. Water-mills unable
to operate led to a lack of flour for baking bread, with inevitable shortages and consequent price rises. Lack of milling capacity meant that the grist was ground rough rather than finer-milled to flour. An attempt at using man-power to turn the mills in Saxony employed 8–10 men, who managed to grind one "*Dresdner Viertel*" (*c*. 27 litres of grain) in two hours (Beyer, 1843). The Dresden administration sent steam engines to its two most important mills, and horse-powered gins
to several others (*Wiener Zeitung*, 18 Sep. 1842). Citing the acute lack of flour, Vincenz Rohn in Litoměřice (Bohemia) obtained permission to build a steam mill there (Katzerowsky, 1895). In Aachen (Germany) and neighbouring parts of Belgium, weaving factories were limited by water shortage and human power was used to maintain at least part of their production (*Allgemeine Preußische Staats-Zeitung*, 22 July 1842).
Dried-up water sources were also indicated by military records, in that they were cited as reasons for the cancellation of manoeuvres, as reported on 20 August from Berlin in Germany (*Wiener Zeitung*, 3 Sep. 1842, p. 1807): "[due to] *incessant heat and a considerable shortage of water,* […] *the consequences of a winter without snow and of a dry spring* […] *wells and small lakes are entirely dry at a majority of settlements and people are having to bring water from great*
*distances.* [This indicates] *that the cavalry can expect to find no watering places;* [since water] *would have to be made available at campsites, manoeuvres were completely cancelled on the evening before muster.*" Similarly, a report from Düsseldorf a day earlier mentioned that if the very hot and dry weather conditions did not change, then the military manoeuvres already taking place would be significantly curtailed (*Wiener Zeitung*, 26 Aug. 1842).
The problems associated with lack of water impact upon animals as well as people. For example, the "Book of Memory" for Heřmanovice (northern Moravia) reports that coachmen took containers of water for their horses from Bohemia to Silesia because most of the wells and streams there were dry (AS3). On 15 June, people in Emden (Germany) gave voice to a host of complaints about a lack of water. The ports and waterways had to be flushed with seawater, so all the city
canals in their  neighbourhood had become salty for up to half an hour and therefore useless for cattle or irrigating the vegetable gardens around the city (*Algemeen Handelsblad*, 26 June 1842).



Small yields of hay and no aftermath (Sect. 4.3.2) led a critical situation for those who raised cattle. A desperate lack of forage meant that farmers were forced to sell off livestock at prices deeply below the normal market rates, or to slaughter livestock for meat. In some cases, livestock numbers dropped to a half, or even a third of previous numbers. This was especially true of cattle
and sheep. Although, whenever possible, farmers found substitute forage (e.g. vetch, tree foliage), the immediate market effects were substantial. For example, the records kept by Václav Křeček from Dobruška (Bohemia) note (Robek, 1978, pp. 49–50): "[…] *a cow, which was* [worth] *100 gulden a year ago, went* [in 1842] *for 40 or 50 gulden, and it was not possible to sell* [a cow] *to anyone because of too little fodder. A two-year heifer could be bought for 20 gulden, maximally 25*
*gulden, while a year ago* [the price for such a beast] *was 50, or even as much as 60 gulden*. […] *Older cows could be bought for 25 or 30 gulden*." In the former Hungarian domain of Levoča (now Slovakia), an irrigation system for meadows and gardens was introduced in 1842 in an attempt to alleviate any future droughts (Érkövy, 1863; see also *Magyar Gazda*, 24 Nov. 1842).

Town and city administrations were also obliged to address the critical scarcity of water. For
example, on 6 June, the council of the city of Bruges (Belgium) forbade the washing or cleaning of the streets, pavements, houses and windows "*until further notice*" (*Gazette van Brugge*, 6 June 1842, p. 2). The mayor and municipal executive of the city of Middelburg, Zeeland in The Netherlands, in consideration of the most economical use of the water, decided to forbid sanding the streets, the pavements and houses, and the washing of windows (*Middelburgsche Courant*, 11 Aug.
1842). In Haarlem (The Netherlands), it was announced on 19 August that the city and neighbourhood pumps will remain closed at night and would be opened for only two hours a day for residents. The inhabitants were furthermore enjoined to use water sparingly, and to use it as little as possible to rinse and scrub the streets and pavements (*Algemeen Handelsblad*, 22 Aug. 1842).

The governor of the province of West Flanders (Belgium) adopted quite drastic measures to
conserve water. He imposed restrictions on the use of the locks on the River Yser and its associated canals. In particular, in consideration of petitions from local authorities concerned at the scarcity of water in the interior, the usual drainage and flushing of the canals was to be delayed from the 1st until the 20th of September (*Gazette van Brugge*, 17 Aug. and 24 Aug. 1842).

To minimise the danger of fire during prolonged dry weather, the mayor and municipal
executive of the city of Rotterdam (The Netherlands) decided that any pyrotechnics, such as aerial candle-lanterns, flares, or fireworks, would not be permitted (*Rotterdamsche Courant*, 9 Aug. 1842). On 17 August, the equivalent city authorities in The Hague decided, bearing in mind the catastrophic consequences of a possible fire, to prevent many city pumps providing sufficient water for scrubbing the streets and similar work (*Dagblad van 's Gravenhage*, 19 Aug. 1842). In
Hungarian Szeged even smoking a pipe was permitted due to the heat and drought (Reizner, 1899).

There was an interesting consequence of the 1842 drought in the Neusiedlersee/Fertő-tó area, which straddles the Austrian-Hungarian border. Low water-levels allowed archaeological explorations that resulted in new "prehistoric" findings in the temporarily dry part of its enormous lake basin (Élő, 1937).
Great heat and drought, particularly in August 1842, contributed to certain illnesses, particularly the respiratory, but also including fevers and malaria, as described by the local medical officer in Uzhorod (Ukraine) in the former Hungarian county of Ung (see Sect. 2.1, point vi).

Social reflections of the 1842 drought may be found in practices of religious institutions such as the organisation of masses, prayers, and formal public processions to entreat God for rain.
For example, from Munich (Germany) on 2 July, regular processions of entreaty for rain are reported from the countryside, as well as one planned also in town in the coming days (*Allgemeine Preußische Staats-Zeitung*, 8 July 1842). On 17 June and on 18 August, the Bishop of Bruges (Belgium) decreed prayers for rain (*Middelburgsche Courant*, 18 June 1842; *Gazette van Brugge*, 19 Aug. 1842). In Antwerp (Belgium), public prayers started on 26 August in all the city's
churches, although some rain had fallen, because the drought remained "fearful" for people and animals. In the neighbourhood of Zandhoven, it was decided that houses should no longer be sanded for fear of shortage of drinking water, as all the wells were dry (*Gazette van Brugge*, 29



Aug. 1842). In Hostín, Bohemia, Josef Vorlíček reported frequent masses and prayers for rain, as well as the occasional procession of entreaty in 1842, but he thought "*that God has blocked his ears and does not hear our voice*" (Robek, 1958b, p. IV/28). Considerably more curious was an explanation offered for the lack of rain in the Mladá Boleslav region (Bohemia): "*It is said that it is*
*not raining because the last time someone died in Plasy, they put a little down blanket under the dear departed's head and said that if the feathers did not rot, it would not rain*" (Kamper, 1927–1928, p. 247).

        More seriously, the drought of 1842 was considered a consequence of the solar eclipse of 8 July (e.g. Heisig, 1929). This led Franz von Paula Gruithuisen (1774–1852), Professor of
Medicine and Astronomy in Munich (Germany), to a statement in *Münchener politische Zeitung* on 2 July, in which, based on an analogy with the drought of 1811, he associated the extreme drought of 1842 with a lack of sunspots: "*There is very widespread opinion that the imminent solar eclipse* [8 July] *is responsible for the great drought of the current spring and summer. Being more closely acquainted with the influence of the Sun on the Earth, I may assert firmly that the recent drought is*
*related to a striking lack of sunspots. The recent weather is similar to that of the year 1811, in which the same* [degree of] *drought and the same lack of the sunspots that usually tend to herald steady weather, occurred, when the same* [degree of] *drought, the same state of the grain and the same splendid growth of grapes* [were observed] […]" (*Österreichischer Beobachter*, 5 July 1842, p. 723). A full solar eclipse was so impressive that in the French *L'Écho de la Frontière*, reporting
for Valenciennes (1 and 2 Aug. 1842, p. 1) appears: "[…] *the year 1842 will be classified among the numbers of years of abundance of all kinds, together with its good wine* [which]*, if The Lord deigns so to do, may be called 'the wine of the eclipse',* [just] *as that of 1811 was named the 'wine of the comet'* [Flaugergue's comet]".

**5 Discussion**
**5.1 The 1842 drought in the longer-term context**
Starting at the western part of the territories analysed herein (Fig. 2), Slonosky (2002) identified 1842 as the tenth driest year in the homogenised precipitation series for Paris in the 1688–2000 period. With its annual total of only 401 mm it was the second driest year of the 19th century, a
little behind 387 mm in 1884. According to seasonal and annual precipitation totals reconstructed from documentary-based indices for the Czech Lands in the 1501–2010 period (Dobrovolný et al., 2015), the 1842 JJA precipitation was the second driest season (just behind the extraordinary year of 1540). The same was found for annual precipitation. Furthermore, documentary-based indices rank 1842 as the driest spring (MAM) in the 1501–1854 period (together with 1571, 1638, 1683,
1779, and 1790).

        In the context of instrumental series, April–September 1842 was the most extreme drought in the Czech Lands in the 1805–2012 period according to SPI-1, while it was the third worst in terms of Z-index and the fourth most severe according to SPEI-1 (Brázdil and Trnka, 2015). Drought chronology created by a combination of documentary and instrumental data indicates that,
for JJA as well as for the summer half-year (April–September), the drought of 1842 had a return period of 200 years for SPI and 50 years for SPEI and Z-index (Brázdil et al., 2019a). The summer of 1842 was the second-driest event according to scPDSI (sc – self-calibrated) in the north-east sub-region of the Greater Alpine Region during the 1800–2003 period (van der Schrier et al., 2007). Inglot (1968) reported the MAM–JJA drought of 1842 as among the most important droughts in
Lower Silesia. A report from Tallin, in Estonia, dating to 15 September 1842, speaks of a sudden onset of mild autumn weather following on from periods of extraordinary heat and continual drought (Tarand et al., 2013).

        Ogrin (2002), in a contribution addressing droughts in the Primorsko region of south-west Slovenia, reported severe drought from July to September 1841 (Trieste) and in August–September
1842; August–September 1843 were also dry. This tallies with historical records from neighbouring Hungary, where droughts were also reported for these three years (with catastrophic hay harvests and bad harvests of most crops, especially in 1841).





In eastern Europe, Russian annals reported drought (*zasucha*) occurring in some regions and bad harvests in the Ekaterinburg, Tavrida, Pskov, Mogilev, Podolsk, Vitebsk and Perm districts. However, a wet summer was reported in the Stavropol region and only August was dry with clear skies. Locust plagues occurred, particularly in the Poltava and Černigov regions (Borisenkov and
Pasetskiy, 1988). In Europe, a report dated 10 May mentions the first swarms of locusts in the province of Messinia (Greece) but they were destroyed by the inhabitants under the direction of the authorities. However, more swarms followed, and a part of the yield was destroyed (*Journal des débats politiques et littéraires*, 6 June 1842).

**5.2 Reflections of the 1842 drought in tree-rings and phenophases**
Many tree-rings in Europe are sensitive to hydroclimatic conditions; it allows reconstruct drought patterns (PAGES Hydro2k Consortium, 2017). According to the Old World Drought Atlas – OWDA (Cook et al., 2015), summer scPDSI values derived from tree-ring widths (TRWs) in Europe (Fig. 9) demonstrate dry patterns for 1842, especially in two south-west to north-east belts:
the first over western Europe from south-western France to the North Sea (quite intense drought) and the second located more to the east, from the Adriatic Sea to south-eastern Poland and western Ukraine. The northern and western parts of the Iberian Peninsula, a greater part of the British Isles, and a part of western Russia also appear dry.

Among individual hydroclimate reconstructions based on TRWs, the year 1842 appears as
the eleventh-driest in terms of May–June Z-index derived from living and historical firs (*Abies alba* Mill.) in Southern Moravia during the 1500–2007 period (Büntgen et al., 2011a). On the other hand, 1842 does not appear among extremes of the fir TRWs sensitive to springtime hydroclimate in the 962–2007 period derived from 11,873 samples from France, Germany, Switzerland and the Czech Lands (Büntgen et al., 2011b). It is also absent, for example, from the 20 most negative JJA
scPDSIs reconstructed for 1744–2006 from 86 pine (*Pinus sylvestris*) TRWs in northern Slovakia (Büntgen et al., 2010).

Droughts, often accompanied by high temperatures, may influence the onset of certain phenophases, bringing them on early. For example, in a report from the Bergstraße (an ancient, *c.* 80-km-long trade route in south-western Germany), dated 12 June 1842 reads: "*Our vineyards*
*are in full bloom; all will have finished in 8 to 10 days, and so in 1842 a rare occasion will arise, that the grape blossoms are over before Saint John's* [24 June]" (*Arnhemse Courant*, 19 June 1842, p. 2). In the vineyards of Kőszeg (Hungary), the grapes started to ripen more than two weeks earlier than usual; however, when the time came to harvest them in October, they were only three days earlier than usual (Kiss et al., 2011). In the neighbouring town of Szombathely the grape harvest
was forced by copious rain to start earlier than planned (5 Oct.). This also affected the quality of the wine, as it probably did in Kőszeg as well, located only *c.* 20 km from Szombathely (AS4). Brázdil et al. (2011) disclosed that early onset of ripeness in sour cherry (*Prunus cerasus*), common blackthorn (*Prunus spinosa*) and blackheart (*Vaccinium myrtillus*) took place in 1842 at the Hradec Králové station (Bohemia) relative to the 1828–1847 period. Moreover, Brázdil et al. (2019b)
showed that air temperatures combined with drought effects (expressed by SPEI) played a significant role in series of crop- and grape-harvest dates in the Czech Lands in the 1517–1542, 1561–1622, 1770–1815 and 1971–2010 periods; only in the generally wet period of 1871–1910 was any significant effect of SPEI absent.

**5.3 The impacts of the 1842 drought in a broader context**
Depending on the part of the year in which dry spells appear, changes of the prices of agricultural products and goods may be indicators of the effects of bad harvests, despite the many other influences upon them (e.g. damage due to other weather extremes, projected grain yields, export and import of grain, reserves, frequency of grain markets, wars, administrative decrees, and
financial speculation – Petráň, 1977).

The influence of drought on the prices of agricultural crops may be best expressed from prices within the year in question, or the one following it. For example, the Saturday market prices



for Opava (Moravian Silesia) were published in *Troppauer Zeitung*. Despite a relatively high number of missing values, considerable influence of the drought upon the prices of wheat, hay and straw resulting from the 1842 drought is well expressed (Table 1). An increase in wheat prices (expressed in gulden/florins – fl. and kreutzer – kr.) from spring to harvest time (by *c*. 1 fl. from 7

fl. 48 kr.) was probably partly related to fear of lower yields due to drought, but these eventually proved less negatively affected, i.e. prices started to decline after harvest (down to as low as 5 fl. 48 kr. in April 1843). Prices of hay (2 fl. 15 kr. at the end of March 1842) rose steadily after a poor haymaking, and then more sharply after failure of the aftermath. From August 1842 onwards, hay prices remained constantly over 4 fl., dropping slightly below this value only at the end of February

1843. After a steady decrease to 3 fl. in August 1843, prices suddenly dipped slightly below 2 fl. in the following month as a result of a good haymaking. The dry conditions in 1842 resulted in lower quantities of straw than usual. This was reflected in a price increase from 12 fl. at the end of March 1842 to 16 fl. at the end of August. Straw prices fell significantly after the 1843 harvest, to as low as 8 fl. 30 kr. in October 1843 (not shown in Table 1).

In the longer term, the effects of the 1842 drought appear relatively negligible. This is evident, for example, in overall grain production in the Czech Lands, which documentation indicates was not very strongly influenced by the drought (see Sect. 4.3.2). Fig. 10 shows only a slight increase in the prices of basic cereals in Prague from a local minimum in 1841 through 1842, achieving a local maximum in 1843 during the 1832–1852 period. However, far worse increases in

grain prices took place in 1846–1847, the result of very bad grain harvests in these two years, as recorded in many Czech documentary sources. Similarly, the market prices of agricultural products in Vienna in 1842 were often somewhat higher than in other years. However, only the prices of legumes (peas, beans, lentils) rose significantly, by 30%–40% over the previous and following years (see Pribram, 1938).

The fact that no particularly outstanding price increases for base products occurred in either Prague or Vienna could result from a number of factors and administrative strategies or decisions. One important influence was the mass storage of grain in royal/regional granaries; such a reserve could buffer the effects of at least one bad harvest (e.g. *Jelenkor*, 3 Sep. 1842). It was also significant that the official market prices of major food products (meat, cereals, etc.) were fixed by

regional authorities. Despite actual market increases, these regulations may have had a positive impact, and again could have buffered some of the immediate effects of the bad and very bad harvest results of a dry year (e.g. *Nemzeti Újság*, 18 Sep. 1842). Moreover, not all regions were affected to the same degree by drought and related natural extremes. There were areas that saw ample harvests in 1842 within the vast and geographically varied Habsburg empire: for example,

the major grain production regions of southern Hungary had a good harvest, and its cereals were exported. Reports in the *Jelenkor* newspaper suggest regular shipping on the Danube towards Vienna all summer and autumn 1842, evidently not disturbed by low water levels. Further, despite less favourable (or even bad) hay harvests at country level, the export of hay from Hungary to Lower Austria (and Vienna) increased significantly in the dry years of 1841 and 1842; this may

well have been due to the increased hay yields in the Neusiedlersee/ Fertő-tó and Hansag/Wasen wetlands during the dry years (Horváth, 2016).

According to the "Book of Memory" kept by the school in Hlinsko, great poverty and hunger afflicted many places in Bohemia, beginning in spring 1843, due to the previous year's drought and consequent shortage of potatoes, which partly rotted in and after storage. Moreover,

people lacked the money to buy those cereals and potatoes that were available. Direct financial support from the Vienna court and public subscriptions were organised in order to assist those who were suffering most, particularly in certain regions of Bohemia (e.g. the Krušné hory Mountains and the Loket and Žatec regions). Emperor Ferdinand I also lent his support to building new roads and maintaining older ones, thus re-establishing some cash-flow among the more deprived social

classes (Adámek, 1908). That the situation was similarly critical on the German side of the Krušné hory Mountains (Erzgebirge) is evident in a record of the words of an unknown farmer, dated to 20 August 1842: "*Since 1816* [the "Year without a Summer"], *the very opposite of 1842, no year*



*has been so tormented with respect to agriculture* […] *as this* one [1842], *and no other has been so horrible for me as the current* [one]." (*Ökonomische Neuigkeiten und Verhandlungen*, p. 790). In contrast, it was important in historical Hungary and Transylvania that the 1842 drought occurred after the similarly dry, perhaps even more severe, dry year of 1841, when shortages, or even famine, developed in those regions most severely afflicted, mainly areas with the worst harvests on the one hand, and those intrinsically deprived and/or with transportation problems of bulk products on the other. Shortages and even famine were reported among the poor, perhaps others, in Bihar, Nógrád and Szabolcs Counties in Hungary (e.g. *Nemzeti Újság*, 21 May and 30 July 1842; *Jelenkor*, 30 June 1842).

**6 Conclusion**

From this analysis of the 1842 drought in Europe (not including the Mediterranean), the following conclusions may be drawn:

(i) Spatial reconstruction and measured totals indicate that precipitation was significantly reduced, particularly in a broad belt extending from France to eastern central Europe. Despite regional differences, dry patterns prevailed mainly in January–February and, with some regionally different degrees of intensity, between April and August.

(ii) The 1842 drought was primarily driven by the precipitation anomaly and only to a far lesser extent exacerbated by an evapotranspiration demand that was higher than usual, as indicated by the three types of drought indices employed herein.

(iii) Mean monthly SLP maps demonstrate the influence of anticyclones or ridges of high pressure upon the precipitation reduction in some areas. The relationships between monthly precipitation totals and circulation indices (NAOI and CEZI) proved rather complicated.

(iv) The general lack of water giving rise to the drought led to a range of impacts on agriculture, forestry, water infrastructure and society at both personal and administrative levels, human responses to which varied. Impacts conveyed through documentary sources were generally more severe than might be expected from the drought extremity calculated from drought indices. However, societal responses managed to reduce possible negative impacts to acceptable levels and avoid further longer-term consequences.

(v) The effects of the 1842 drought on vegetation derived from documentary data are also clearly reflected in JJA scPDSI reconstructed from TRWs in OWDA (Cook et al., 2015).

(vi) This contribution confirms the importance of combining information from systematic meteorological (hydrological) measurements and documentary evidence for comprehensive description of any dry event, together with its spatiotemporal aspects, synoptic reasons, environmental and societal impacts, and human responses.

**Data availability.** The sources of basic datasets are quoted on the corresponding places of the manuscript. Other data used are available from the corresponding author.

**Author contribution**. The individual authors contributed by documentary data, instrumental records, basic analyses and by creating and discussing the manuscript.

**Competing interests.** The authors declare that they have no conflict of interest.

**Acknowledgements**
R.B., P.D. and L.D. acknowledge financial support from the Czech Science Foundation for project ref. 17-10026S. M.T., L.Ř. and P.Z. were supported by the SustES – Adaptation strategies for sustainable ecosystem services and food security under adverse environmental conditions project, no. CZ.02.1.01/0.0/0.0/16_019/0000797. Andrea Kiss would like to acknowledge financial support from the Austrian Science Funds project ref. I 3174. Jan Balek (Brno) is acknowledged for calculation of drought indices for Fig. 5; Neil Macdonald (Liverpool), Anders Moberg (Stockholm), Christian Rohr (Bern) and Victoria C. Slonosky (Montreal) for help with finding/obtaining





instrumental meteorological series and adding any further information. Tony Long (Svinošice) helped work up the English.

**Archival sources**

[AS1] Csongrád County Archives, Archives of Szeged Town IV.B. 1017, Jegyző-Könyv A' Tisza Vize Tükrének I-ső Januáriustól 31-ik Decemberig 1842-ik Esztendei állapotjáról (manuscript).
[AS2] Dagboek van Doeke Wijgers Hellema at Wurdum, Friesland, The Netherlands, http://www.erfgoed-fundaasje.nl/hellema/dagboeken-van-doeke-wijgers-hellema/dagboek-16-1842/.
[AS3] Státní okresní archiv Bruntál, fond obce Heřmanovice, Pamětní kniha obce Heřmanovice.
[AS4] Vas County Archives, Vas megyei levéltár V/102a, Szombathely város tanács közgyűlési jegyzőkönyvei (Meeting protocols of the council of Szombathely town).

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



Table 1. Saturday's market prices of wheat (*Metzen* – 51.487 l), hay (*Zentner* – 56.006 kg) and straw (60 shocks) in Opava (Moravian Silesia) on selected dates in 1842 and 1843. Prices are expressed in gulden (*Florin* – fl.) and kreutzer (*Kreutzer* – kr.) of Vienna currency. Source of data: *Troppauer Zeitung*

| Date | Wheat | | Hay | | Straw | |
|---|---|---|---|---|---|---|
| | fl. | kr. | fl. | kr. | fl. | kr. |
| 26 March 1842 | 7 | 48 | 2 | 15 | 12 | 0 |
| 28 May 1842 | 8 | 27 | 2 | 24 | 13 | 30 |
| 18 June 1842 | 8 | 42 | 2 | 24 | 14 | 0 |
| 30 July 1842 | 8 | 57 | 3 | 6 | 15 | 30 |
| 27 August 1842 | 7 | 0 | 4 | 24 | 16 | 0 |
| 1 October 1842 | 6 | 48 | 4 | 15 | 15 | 0 |
| 26 November 1842 | 6 | 6 | 4 | 12 | 16 | 0 |
| 14 January 1843 | 5 | 57 | 4 | 9 | 16 | 0 |
| 25 February 1843 | 5 | 54 | 3 | 57 | 14 | 30 |
| 15 April 1843 | 5 | 48 | 3 | 39 | 15 | 0 |
| 1 July 1843 | 6 | 33 | 3 | 18 | 15 | 15 |
| 23 September 1843 | 8 | 12 | 1 | 57 | 9 | - |





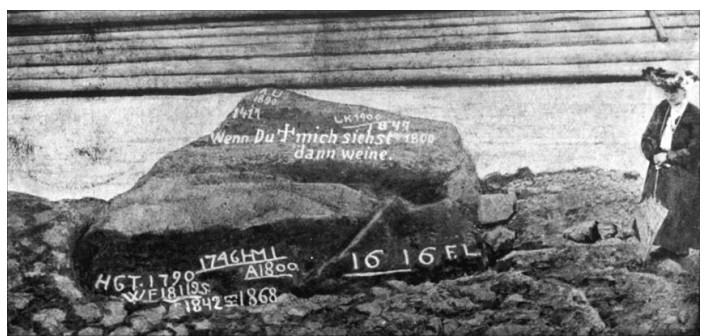

**Figure 1.** Hydrological drought of 1842 as indicated by a mark on a "hunger stone" on the River Elbe in Děčín–Podmokly (Bohemia). This stone appeared during a severe drought in 1904, when the picture was taken (O. Kotyza Archives)





**Figure 2.** Precipitation patterns of DJF (a), MAM (c), JJA (e) and DJF–JJA (g) and their expression as percentages of the corresponding parts of the seasonal means in the 1961–1990 reference period (b, d, f, h) over the European lands (data from Pauling et al., 2006)

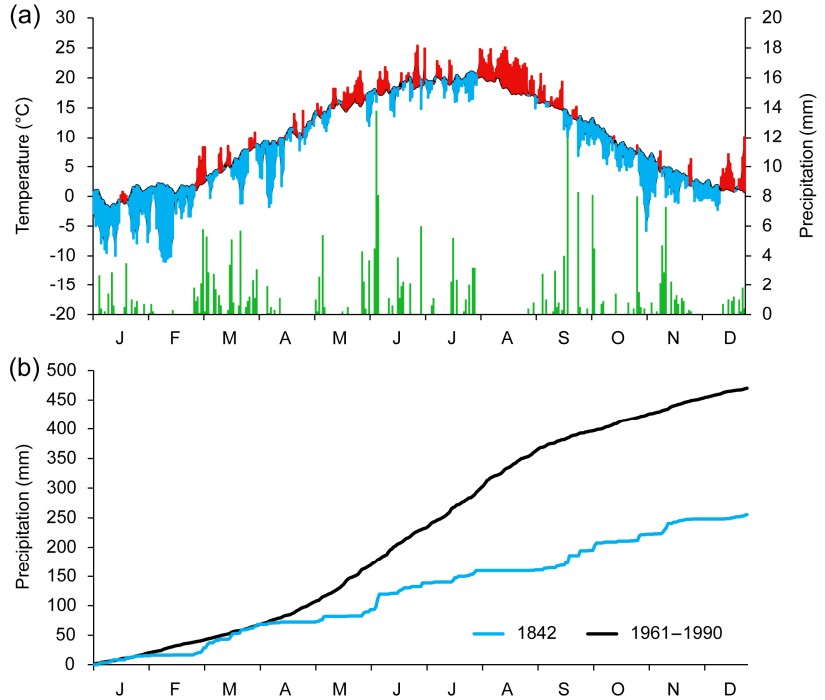

**Figure 3.** Temperature and precipitation for 1842 at the Prague-Klementinum station: (a) annual variation of daily mean temperatures (compared with the 1961–1990 mean variation) and daily precipitation totals for 1842; (b) cumulative daily precipitation totals for 1842 in comparison with the 1961–1990 reference period



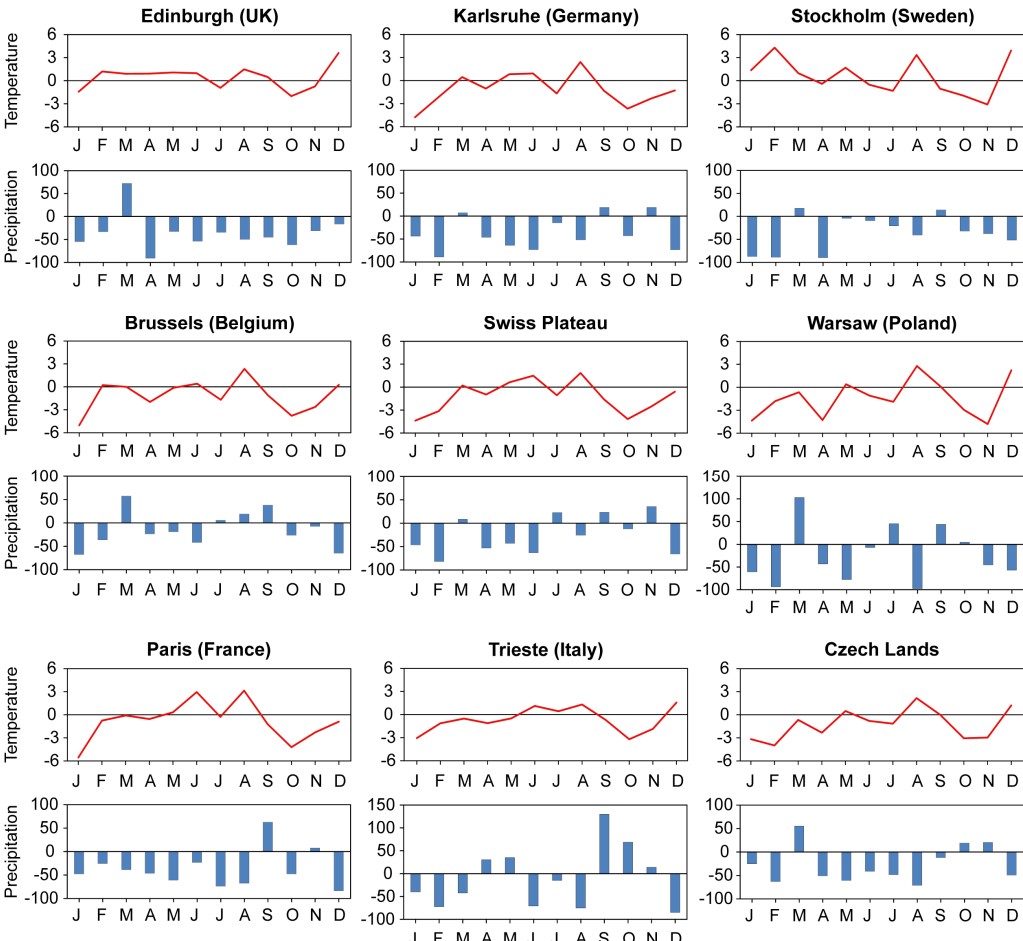

**Figure 4.** Annual variation of air temperature and precipitation in 1842 for selected European meteorological stations and regions. Temperatures are expressed as deviations from the 1961–1990 reference period, precipitation totals as percentages of corresponding 30-year means





**Figure 5.** Annual variation of SPI-1, SPEI-1 and Z-index for selected European stations and regions in 1842







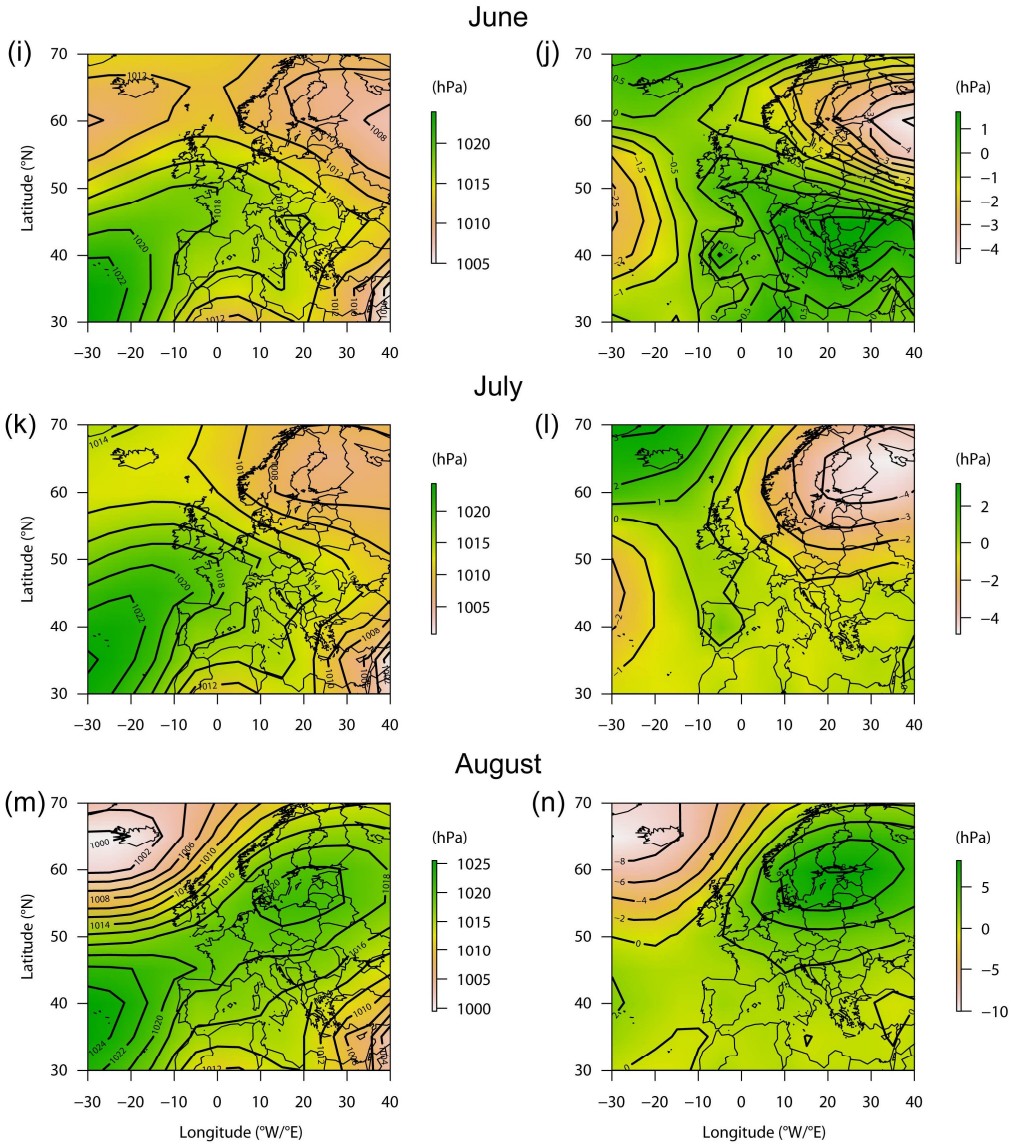

**Figure 6.** Mean sea-level pressure and deviations with respect to the reference 1961–1990 period means in the European-Atlantic area for the months of January (a,b), February (c,d), April (e,f), May (g,h), June (i,j), July (k,l) and August (m,n) 1842



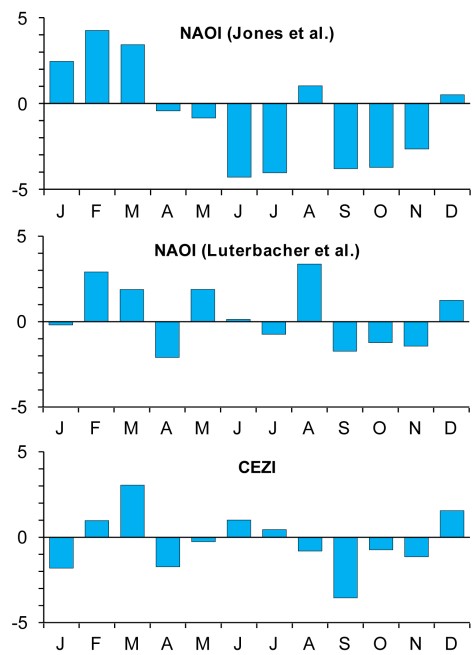

**Figure 7.** Annual variation of mean monthly NAOI (Jones et al., 1997; Luterbacher et al., 1999) and CEZI (Jacobeit et al., 2001) in 1842



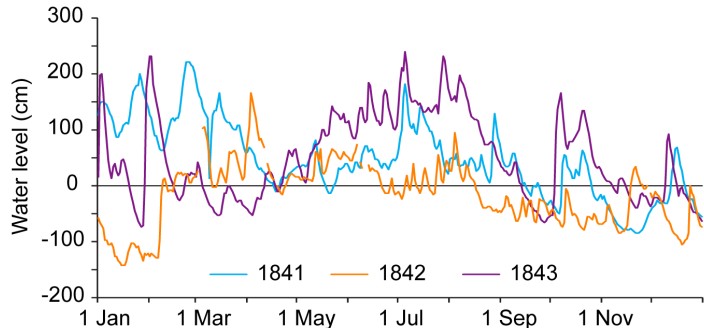

**Figure 8.** Annual fluctuations in daily water levels of the Danube at Vienna in the profile of the
Great Danube Bridge during the years 1841, 1842 and 1843 (data extracted from *Wiener Zeitung*)



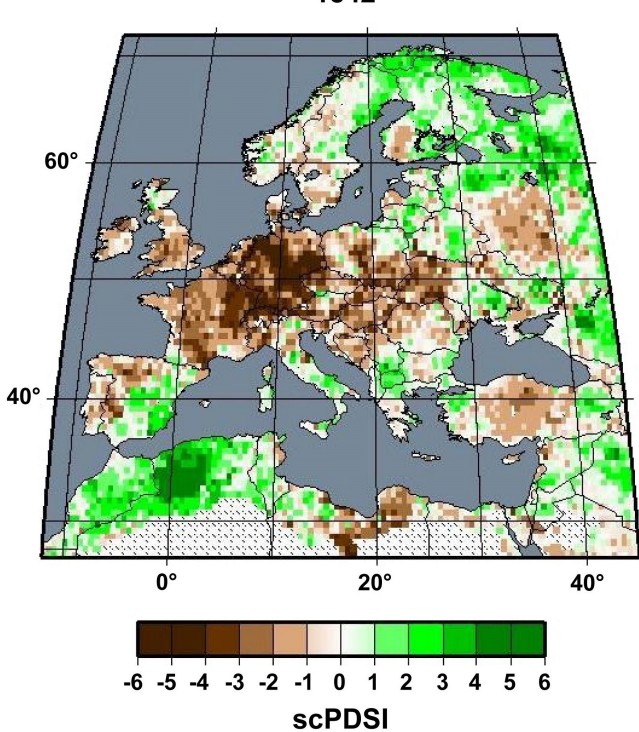

**Figure 9.** Values of summer scPDSI for 1842 in the European area as derived from tree-rings in OWDA (Cook et al., 2015)



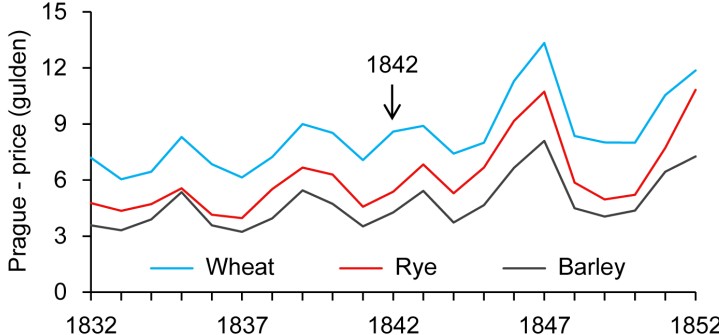

**Figure 10.** Fluctuations in grain prices (wheat, rye, barley) in Prague during the 1832–1852 period
(data from Schebek, 1783)