# Peer review of "The extreme drought of 1842 in Europe as described by both documentary data and instrumental measurements"

_Climate of the Past, 2019_

## Referee Comment (RC1) · Gerard van der Schrier (Referee) · 10 Jul 2019

Review of 'The extreme drought of 1842 in Europe as described in Europe by both documentary data and instrumental measurements' by Brázdil, Demarée, Kiss, Dobrovolny, Chromá, Trnka, Dolák, Reznickova, Zahradnicek, Limanowka and Jourdain.

The paper describes in great detail the drought of 1842 in Europe, both from the perspective of instrumental observations as well as from documentary data. This results in a comprehensive paper on this drought where the spatial extent, temporal characteristics and relation to the atmospheric circulation are discussed and many of the details and impacts are highlighted.

[Figure]

This reviewer is much impressed by this study - the research efforts that have gone into this paper in finding the documentary data is stunning. Although the use of newspaper articles and diary entries are not used for the first time in connection with observational data to produce a view on a particular climatic extreme, it is still a very novel approach and the current paper will probably set a standard. However, there are some aspects of this study that require further work. As the background of this reviewer is in climatology and observational data, much of this work relates to the instrumental data and the relation between the drought and atmospheric circulation, and to place the findings for 1842 in a broader (historical) perspective. The motivation for this study and its relevance for the community need to be highlighted a little better. Finally, there are a few smaller issues related to the use of documentary data and the conclusions than can be drawn based on these type of data.

The group of Kerstin Stahl and Veit Blauhut (University of Freiburg) and Lena Tallaksen (University of Oslo) have done much work on classifying impacts of drought, which might be helpful in presenting the impacts of the 1842 drought is a more systematic manner.

Overall, this is a paper that deserves to be published and my advise to the editor is to accept conditional one some changes.

Major issues

*) Introduction: I understand that 1842 is interesting because of the possibility to combine instrumental and documentary data sources. But why is 1842 interesting (and not another dry year)? What are we likely to learn in the interpretation of modern droughts? Can you link any societal issues or changes to the impact of the 1842 drought so that we might understand societal changes which happened in response to this drought better?

*) page 5, 1st paragraph. The authors are right in noting that systematic efforts to homogenise data - especially daily data - do not really exist. For Europe, there are new

efforts (Squintu et al. 2019a; 2019b) but it is unlikely that this will give homogeneity adjustments for as early as 1842 (because of the lack of reference data for this early period). However, even daily data that is not homogenized will be useful. Below is a summary of daily data that is available in ECA&D for 1842, and it would add to the importance of the study to include these daily data, along with the daily data of Prague, in favour of the monthly (less informative) data now included. ECA&D staff is more than happy to provide these data (under restrictions). Some of these instrumental data might be used to validate the documentary sources, since it is a little strange that e.g the Wiener Zeitung is used as a source for Scottish precipitation (and not a UK paper) (page 7, line 32).

*) page 7, lines 29-42. I am a little uneasy about using reports in newspapers on meteorological observations. In my view, newspaper reports, and other documentary evidence, is a perfect source for impact information. There are many fine examples of this in the current paper. A discussion of meteorological observations in a newspaper, which is not impact information, should be accompanied by a brief analysis of actual observations to back these claims. We are, after all, climatologists.

*) page 16, 1st paragraph and figure 8 and figure 3 (lower figure) A historical perspective is needed here. To appreciate the prices or cumulative precip totals or water levels, you need to make clear that the 1842 values were outside the range of what can be expected to be 'normal'. Give, if data permits, the 5th percentile of driest years in some reference period or (for prices) index the prices using a 'common' year.

Other issues the authors may want to look at

*) page 6, line 9: would it be a possibility to have electronic Supplementary Data which might make these information available?

*) page 6, line 25: Both the SPEI and Z-index require an estimate of the potential evapotranspiration. I take that the Thornthwaite formula is used, but this is not specified. Also, these indices require a calibration period. What period is used? This is not

specified (and will strongly affect results).

*) page 7, line 22. Using the SDII (Simple Daily Intensity Index) might be useful here?

*) page 7, line 42. Do they really lack sufficient systematic character to cite? Perhaps this is true, but it likely reports on the most extreme situations, not only related to local conditions but also in the eyes of the journal editor. This will indeed give a randomness in reporting, but in this randomness, the underlying phenomenon may have spatial consistency. These reports give, at least, an indication of the spatial extent of the most extreme situation.

*) section 4.2: Can you comment on similarities between these SLP patterns, where high pressure seems a constant factor, and other (modern) droughts? Or to put it differently, is the circulation which led to the 1842 drought a unique one-off situation or do you see some similarities between the driver of this drought and other droughts?

*) page 9, line 17. Perhaps a little more information is needed here on the zero point. Is that also a supposed minimum as in an earlier example, or some other reference?

*) page 9, line 42-48. These islands in the southwest of the Netherlands are surrounded by sea water and have no rivers or freshwater lakes. This makes them vulnerable to drought. The situation is likely to be worse there than for other parts of the Netherlands.

*) page 11, line 19-26 and page 15, line 1-8. The relation to drought is not clear - there is only a claim in the newspaper. The drought and pests plague may be coincidental without the drought being a cause. Please present independent evidence that drought relates to the outbreaks of these worms/insects.

*) page 12, line 49-51. I'm confused what this report actually means, apparently it is salt water intrusion in the waterways which will make the water useless for irrigation or drinking. But for a period of half an hour? I'm not sure if that is even possible.

*) page 13-14, lines 43-23. I don't see why this is added to the text. It does not clarify

the impact of the drought, it only gives information on the science of that period (which is not the topic of the paper).

\*) page 14, lines 36-43. This text requires a little more interpretation. The difference between the ranking based on SPI and SPEI is confusing otherwise. Clearly, the amount of precipitation was very low (making the SPI very negative), but the temperatures were not that high (especially in winter and spring),making the estimate you make for PET relatively low. This is the only reason why the SPEI gives such mundane values. You see that the scPDSI, which is based on a water balance (in contrast to the SPEI) does not give as much weight to the PET value as the SPEI (it calculates the actual evapotranspiration) and gives a drought ranking that is more in-line with the SPI value.

\*) page 15, line 19-26. here additional interpretation is needed as well, why do these trees not pick-up the drought signal? Are they living is conditions which are temperature stressed rather than moisture stressed?

\*) page 16, line 25-41. The increase in the slaughter of cattle because farmers are unable to provide fodder or water, must have decreased the price of meat - any clues for that?

Very very minor issues

\*) page 2, line 19. Would 'Shortage' be a more appropriate translation than 'Poverty'?

\*) page 13, line 35: I would expect that it was NOT permitted to smoke a pipe....

\*) caption fig 10, the year should be 1873

\*) caption fig. 4, These are Monthly variations (not annual)

Squintu, A.A. et al. (2019a), doi:10.1002/joc.5874

Squintu, A.A. et al. (2019b), Building long and homogenous temperature series in ECA&D: blending of neighbour series, J. Applied Meteorology and Climatol. (submitted).

[Figure]

| 17 | Uccle | be | tg | 1833-01-01 | 2008-01-31 |

| 17 | Uccle | be | tn | 1833-01-02 | 2011-01-23 |

| 17 | Uccle | be | tx | 1833-01-06 | 2011-09-30 |

| 27 | Praha-Klementinum | cz | rr | 1804-05-01 | 2005-04-30 |

| 27 | Praha-Klementinum | cz | tg | 1775-01-01 | 2005-04-30 |

| 27 | Praha-Klementinum | cz | tn | 1775-01-01 | 2005-04-30 |

| 27 | Praha-Klementinum | cz | tx | 1775-01-01 | 2005-04-30 |

| 11744 | Dresden (Mitte) | de | rr | 1828-01-01 | 1992-11-30 |

| 11744 | Dresden (Mitte) | de | tg | 1828-01-01 | 1915-12-31 |

| 11744 | Dresden (Mitte) | de | tn | 1828-01-01 | 1915-12-31 |

| 11744 | Dresden (Mitte) | de | tx | 1828-01-01 | 1915-12-31 |

| 48 | Hohenpeissenberg | de | rr | 1781-01-01 | 2019-06-30 |

| 48 | Hohenpeissenberg | de | tg | 1781-01-01 | 2019-06-30 |

| 49 | Jena Sternwarte | de | hu | 1824-11-20 | 2019-06-30 |

| 49 | Jena Sternwarte | de | rr | 1826-12-01 | 2019-06-30 | | 49 | Jena Sternwarte | de | tg | 1824-01-01 | 2019-06-30 |

| 49 | Jena Sternwarte | de | tn | 1824-01-01 | 2019-06-30 | | 49 | Jena Sternwarte | de | tx | 1824-01-01 | 2019-06-30 |

| 28 | HELSINKI KAISANIEMI | fi | tg | 1828-10-04 | 2001-12-31 | | 271 | Armagh | gb | hu | 1838-01-02 | 2008-12-31 | | 271 | Armagh | gb | rr | 1838-01-01 | 2001-12-31 |

| 257 | CET Central England | gb | tg | 1772-01-01 | 2003-05-31 |

| 274 | Radcliffe Meteorological Station Oxford | gb | rr | 1827-01-01 | 2019-04-30 |

| 274 | Radcliffe Meteorological Station Oxford | gb | tg | 1815-01-01 | 2019-04-30 |

| 274 | Radcliffe Meteorological Station Oxford | gb | tn | 1815-01-01 | 2019-04-30 |

| 274 | Radcliffe Meteorological Station Oxford | gb | tx | 1814-12-31 | 2019-04-29 |

| 169 | Bologna | it | rr | 1813-01-01 | 2007-12-31 |

| 169 | Bologna | it | tg | 1814-01-01 | 2003-12-31 |

| 169 | Bologna | it | tn | 1814-01-01 | 2003-12-31 |

| 169 | Bologna | it | tx | 1814-01-01 | 2003-12-31 |

| 171 | Genoa | it | rr | 1833-01-01 | 2008-12-31 |

| 172 | Mantova | it | rr | 1840-04-01 | 2008-12-31 |

| 173 | Milan | it | tg | 1763-01-01 | 2008-11-30 |

| 173 | Milan | it | tn | 1763-01-01 | 2008-11-30 |

| 173 | Milan | it | tx | 1763-01-01 | 2008-11-30 |

| 380 | Padova | it | tg | 1725-01-12 | 1997-05-31 |

| 380 | Padova | it | tn | 1774-01-01 | 1997-05-31 |

| 380 | Padova | it | tx | 1774-01-01 | 1997-05-31 |

| 381 | Palermo | it | rr | 1797-01-01 | 2008-12-31 |

| 85 | St. Petersburg | ru | tg | 1743-03-04 | 1999-08-31 |

| 10 | Stockholm | se | tg | 1756-03-27 | 2003-12-31 |

| 426 | Uppsala | se | rr | 1840-01-01 | 2001-12-31 |

| 426 | Uppsala | se | tg | 1840-01-01 | 2001-12-31 |

| 426 | Uppsala | se | tn | 1840-01-01 | 2001-12-31 |

| 426 | Uppsala | se | tx | 1840-01-01 | 2001-12-31 |

---

## Referee Comment (RC2) · Anonymous Referee #2 · 22 Jul 2019

The extreme drought of 1842 in Europe as described by both documentary data and instrumental measurements.

This is overall a nice paper that sheds light on an important European drought event. The paper is well written and presented and does a good job of illustrating the power of bringing together both qualitative and quantitative data to understand an important historical event. That said, I have some comments that need to be addressed by the authors.

The most significant comment I have is around the structure of the paper. It seems that results are presented throughout the paper including in the introduction which highlights some of the documentary impacts of the 1842 drought, and in the discussion where new results are presented to the reader. In particular I would interpret section 5.1 and 5.2 as results. I will leave it to the editor to decide this but I would prefer to see these integrated into results as the paper is using documentary and instrumental data. If the authors would prefer to have the proxy tree ring data as part of the discussion then I can understand that. The role of the discussion section should be to discuss the results and place them in a broader spatial/temporal context and to discuss any limitation, assumptions etc that were part of the analysis. The latter in particular could be fleshed out a bit more than is presently the case. Taken together with other comments below I feel that the outcome should be accept with minor revision as little new analysis would be required.

Other comments Acronyms are used in the abstract, while some like NAO are widely known others like CEZI, SPI, SPEI, Z-index may not be, please spell these out for the reader.

It would be useful to have a map of Europe showing from where instrumental and documentary sources are derived from. This would help convey the continental nature of this event and its impacts.

In describing the Pauling et al data please give a references for the gridded analysis from 1901-2000.

Does the Pauling et al data include precipitation from the individual series that you present later from across Europe and if so does this introduce a circularity into using these as independent pieces of information to assess the magnitude of the drought?

In addition, perhaps I missed it but in the data section an overview of the precipitation gauges used later in the paper is not provided. In addition are there other series and regional precipitation records that might be usefully used to extend the quantitative assessment?

Why did you use SPI/SPEI 1 and not longer accumulations given that much of the focus is on agricultural, hydrological and socio-economic drought. Some thoughts on this either in the methods or in the discussion would be welcome. Does the result change if you do?

More detail is needed on how the drought indicators were derived, just saying that 'These series were then worked up' does not allow the work to be repeated.

Has the homogeneity of the various instrumental records used been assessed? If so/not this needs to be stated and if necessary returned to in the discussion. This is an early period in the observational history and gauges and their exposure often very different that today. More comment is needed on this.

In terms of the consideration of hydrological drought why not look at 1842 in the context of the long term mean as you have done with precipitation? Only two adjacent years are used. Is the data not available? It seems it is from what is presented.

Use of documentary sources is very good and indeed a standard to be aspired to.

See points above on discussion where I think most work is needed in revising.

---

## Referee Comment (RC3) · Anonymous Referee #3 · 4 Aug 2019

This is a historical climatological work of high quality based on an impressive amount of data and I recommend it to be published after minor revisions.

In particular, it conforms to the current trend of combining different types of sources. Such source pluralism though, especially when the object of study belongs to modern times, entails rather superficial or nonexistent assessments of the quality of the sources, particularly the documentary evidence. Some words about the problems connected to such anthropogenic sources would be appropriate. The present disposition of the article also allows for that in a pedagogical sense; the documentary sources (2.1 Documentary data) are presented first under 2 Data, and the not unequivocal reliability and representativity of these makes the following presentation of instrumental data the more relevant since the complementarity and usefulness of both source categories become clear.

The discussion about prices (pp 16f) is interesting but also highlights the problems connected to the use of prices to shed light upon climatic extremes. The authors are rightly cautious but the problems could be stressed even more. The climatological signal from agricultural prices is hard to detect without identifying and eliminating the 'noise' in the data and depends on whether market mechanisms are allowed to operate freely, the degree of long-distance market integration etc, all of which is especially true for pre-industrial Europe when market demand for basic foodstuffs varied very little in the short run in contrast to supply. Some of these aspects are indeed mentioned in the present article and cannot be fully addressed there. But I would suggest that the clearly more significant importance of agricultural yields over agricultural prices should be recognized.

Finally, some small linguistic details:

Page 2, line 3: "droughts events" should be replaced by "drought events"

Page 8, line 35: "positives modes" should be replaced by "positive modes"

Page 13, line 1: "led a critical situation" should be replaced by "led to a critical situation"

Page 13, line 35: "permitted" should be replaced by "prohibited" (or something similar)

---

## Author Comment (AC1) · 24 Aug 2019

Gerard van der Schrier (Referee) schrier@knmi.nl Review of 'The extreme drought of 1842 in Europe as described in Europe by both documentary data and instrumental measurements' by Brázdil, Demarée, Kiss, Dobrovolny, Chromá, Trnka, Dolák, Reznickova, Zahradnicek, Limanowka and Jourdain.

The paper describes in great detail the drought of 1842 in Europe, both from the perspective of instrumental observations as well as from documentary data. This results

in a comprehensive paper on this drought where the spatial extent, temporal character- istics and relation to the atmospheric circulation are discussed and many of the details and impacts are highlighted.

This reviewer is much impressed by this study - the research efforts that have gone into this paper in finding the documentary data is stunning. Although the use of newspaper articles and diary entries are not used for the first time in connection with observational data to produce a view on a particular climatic extreme, it is still a very novel approach and the current paper will probably set a standard. However, there are some aspects of this study that require further work. As the background of this reviewer is in climatology and observational data, much of this work relates to the instrumental data and the relation between the drought and atmospheric circulation, and to place the findings for 1842 in a broader (historical) perspective. The motivation for this study and its relevance for the community need to be highlighted a little better. Finally, there are a few smaller issues related to the use of documentary data and the conclusions than can be drawn based on these type of data. The group of Kerstin Stahl and Veit Blauhut (University of Freiburg) and Lena Tallaksen (University of Oslo) have done much work on classifying impacts of drought, which might be helpful in presenting the impacts of the 1842 drought is a more systematic manner. Overall, this is a paper that deserves to be published and my advise to the editor is to accept conditional one some changes. RESPONSE: We thank the referee for careful reviewing of the paper and generally positive evaluation. As for the motivation of this study, we believe that it was explained well in the Introduction. On page 2, lines 15-29 is clearly stated that the 1842 drought was evaluated as exceptional by contemporary sources and it was reported also in several other Central European papers based on documentary as well as instrumental meteorological data. With the fact, that this event is well covered by documentary and instrumental data, these facts are a good reason to study this event in the broader European scale. We fully respect the works of the group of Kerstin Stahl and Veit Blauhut (University of Freiburg) and Lena Tallaksen (University of Oslo), but we do not understand how their classification of impacts would be useful for just one year, if we do

not make comparison with impacts of droughts in other years. We believe that dividing of impacts into different sectors after types of drought on hydrological, agricultural and socio-economic droughts gives well the idea about the importance of corresponding drought impacts, further discussed in Section 5.3.

Major issues *) Introduction: I understand that 1842 is interesting because of the possibility to com-bine instrumental and documentary data sources. But why is 1842 interesting (and not another dry year)? What are we likely to learn in the interpretation of modern droughts? Can you link any societal issues or changes to the impact of the 1842 drought so that we might understand societal changes which happened in response to this drought better? RESPONSE: We believe, that motivation for the study of 1842 drought we explained in the third paragraph of the Introduction, besides the possibility to analyse a drought event quite well covered by documentary and instrumental data. We also have to take in account that not all extraordinary droughts have disastrous effects on the society as a whole or end up as a catastrophe: the severity of societal impacts significantly depends on the preparedness of society, the individuals, and particularly the state and territorial/local administration as well as the trade. The 1842 drought is one of the outstanding drought episodes that cover extensive areas of Europe in a transitional period, when already quite reliable instrumental observations are available, but society and societal reactions still reflect on the response of a pre-industrial society. By the early 1840s, this part of Europe was much more prepared (with regional state/regional or urban grain storages, distribution system, major trade routes etc.). This is a major difference from some other regions of Europe, for example, in the north (e.g. Finland). Moreover, in the light of the widespread crisis in the second half of the decade, the impacts of this drought event alone are, however, widespread and important, not outstanding. In general, society of the study period and the affected western and central-European region was relatively well-prepared for one-one worse harvests caused by weather extremes – unless they lasted for many years. In our presented case we can talk about one (or two) year of extreme. Moreover, as some of the areas were not so much affected by the drought problem (e.g. those under significant

Mediterranean or eastern influence: e.g. the south of historical Hungary, Croatia, south of France; Poland or western Ukraine, respectively), at least in the cities (along major trade routes) even inland trade could mostly solve food-supply problems (e.g. within the Habsburg Empire), and the bad harvest was only followed by a moderate increase of prices. Greater increase in prices occurred in those areas with bad harvest, worse trade connections. Thus, a particularly important point regarding social impacts is the relatively moderate socio-economic impact, namely that the society in these years was mainly sufficiently prepared (i.e. food reserves, trade from good-harvest areas) so that the negative impacts of this great drought are only lightly visible, for example, in crop prices. In this period mainly the poorest part of society suffered. In conclusion, we do not think that one has to discuss in a journal of broad audience only those single drought extremes that had outstanding impacts on the society; in this way, we would lose the cases and periods (and regions) when and where local and regional societies could cope with an extensive drought in a relatively successful way and, therefore, no major adjustments were needed. Thus, not only those cases should be considered when societies failed to cope with large-scale weather extremes such as drought, but also the positive examples, which was the result of previous development in coping strategies, on local and societal level, should be also considered and presented. We believe that this case is a very good example for that.

\*) page 5, 1st paragraph. The authors are right in noting that systematic efforts to homogenise data - especially daily data - do not really exist. For Europe, there are new efforts (Squintu et al. 2019a; 2019b) but it is unlikely that this will give homogeneity adjustments for as early as 1842 (because of the lack of reference data for this early period). However, even daily data that is not homogenized will be useful. Below is a summary of daily data that is available in ECA&D for 1842, and it would add to the importance of the study to include these daily data, along with the daily data of Prague, in favour of the monthly (less informative) data now included. ECA&D staff is more than happy to provide these data (under restrictions). Some of these instrumental data might be used to validate the documentary sources, since it is a little strange that e.g

the Wiener Zeitung is used as a source for Scottish precipitation (and not a UK paper) (page 7, line 32). RESPONSE: The new homogenisation efforts have been included into the manuscript quoting the both papers by Squintu et al. (2019a, 2019b). Many thanks for the kind offer to use other data, including daily records, from ECA&D for 1842. But we would like to stress that we are trying to keep any acceptable size of the manuscript on the one hand and some balance among individual parts of the paper on the other. It means, that the daily data of Prague-Klementinum were used as some example from territory where meteorological drought was quite severe (e.g. according to SPI with return period 200 year) besides the fact that this data are quality checked, homogenised and allow put them into historical context (the 1961–1990 reference period). From the above reasons we do not see as profitable to include other examples. By the way, all stations given by you at the end of the review we took in account on the base of monthly temperature and precipitation values. Concerning of the Wiener Zeitung: It is very often case that some newspaper reprinted information from other newspapers or gave information from their correspondents in abroad. There is physically and technically impossible to look on all local sources of the similar character and check such information there because local sources are simply not available for us (e.g. they are not digitised or accessible). By the way, this particular case of very low April precipitation total is confirmed by many precipitation series from that area we considered, it means that it should be taken as very realistic statement.

*) page 7, lines 29-42. I am a little uneasy about using reports in newspapers on meteorological observations. In my view, newspaper reports, and other documentary evidence, is a perfect source for impact information. There are many fine examples of this in the current paper. A discussion of meteorological observations in a newspaper, which is not impact information, should be accompanied by a brief analysis of actual observations to back these claims. We are, after all, climatologists. RESPONSE: In this paragraph we are just showing some examples of meteorological information included in different newspapers. We are not using it for any other purposes or formulation of any conclusions. We understand the referee that he would like to see "analysis of

actual observations" but what it brings to the paper? We are not here analysing any reliability of newspaper weather reports. Moreover, even a "brief analysis" of each of these observations in such an overview paper would modify negatively the proportions of the paper that, we think, it is not useful in this case.

*) page 16, 1st paragraph and figure 8 and figure 3 (lower figure) A historical perspective is needed here. To appreciate the prices or cumulative precip totals or water levels, you need to make clear that the 1842 values were outside the range of what can be expected to be 'normal'. Give, if data permits, the 5th percentile of driest years in some reference period or (for prices) index the prices using a 'common' year. RESPONSE: Page 16, the first paragraph: We understand the referee's comment but the aim of this paragraph is to show one example of changes in prices during the 1842-1843 years. There is not easy to find such "normal" price rates for each series, because also economic historians who have much higher expertise in these matters than our team, study more the relative fluctuations, changes between years or periods, and usually do not give a "fix" price. "Fixed" thinks are the logic of a climatologist, but not of a historian, who think in the dynamics of matters, and not in "fixed" values. We believe, that it represents in the context of this article a useful additional information for those who would further like to study the question, in more (local or regional) details. Figure 8: This figure should document rather relative changes of water levels in 1842 compared to two neighbour years of 1841 and 1843. We are very sorry, but we do not have any other daily data to follow the referee's comment. Figure 3b: The figure was changed as requested by adding a band representing the inter-quartile range as well as 5% percentile of the driest years in the 1961-1990 reference period. In 1842, the annual precipitation sum in Prague-Klementinum was the lowest during the period of measurements. The figure caption was changed accordingly.

Other issues the authors may want to look at *) page 6, line 9: would it be a possibility to have electronic Supplementary Data which might make these information available? RESPONSE: We are sorry, but this issue is very problematic. It included many thousands of pages of extracted reports from English, French, Dutch, Belgian, German, Swiss, Austrian, Czech, Polish and Hungarian newspapers and published material. Besides the fact that in local languages written reports would be not understandable to many of potential readers, publication of such material with good grammar would need a great additional investment of time and effort for which we do not have any capacity. We think that to the understanding of the many aspects of this drought, discussed in the paper, we provided a sufficient number and types of examples. It would not add much to the present understanding to publish all original local records, often repeating similar types of information. If anybody will be interested in any concrete particular information, there will be no problem to provide him/her with such report.

*) page 6, line 25: Both the SPEI and Z-index require an estimate of the potential evapotranspiration. I take that the Thornthwaite formula is used, but this is not specified. Also, these indices require a calibration period. What period is used? This is not specified (and will strongly affect results). RESPONSE: The Thornthwaite formula has been used as the available data (monthly mean temperature and precipitation) has not allowed for use of other more robust methods. However, as we are studying primarily period not strongly affected by the anthropogenic climate change, we are not comparing the absolute values of the indices between sites or years but rather studying if 1842 appears to show some signs of drought we think the use of indices provides valuable insight despite less than perfect method used for estimating potential evapotranspiration. As for the reference period we strictly used 1901-2000. Both information is now prominently mentioned in the text.

*) page 7, line 22. Using the SDII (Simple Daily Intensity Index) might be useful here? RESPONSE: Indeed use of SDII would be of a great value unfortunately daily precipitation records have so far not been made available from large majority of sites and providing the information only for few of them, for which data are available with unknown quality/homogeneity, would not help the paper's coherency.

*) page 7, line 42. Do they really lack sufficient systematic character to cite? Perhaps

this is true, but it likely reports on the most extreme situations, not only related to local conditions but also in the eyes of the journal editor. This will indeed give a randomness in reporting, but in this randomness, the underlying phenomenon may have spatial consistency. These reports give, at least, an indication of the spatial extent of the most extreme situation. RESPONSE: The referee is right that it would be possible to detect systematic character, but it is beyond the scope of the paper. This would need huge extra work and will not significantly improve findings in this article.

*) section 4.2: Can you comment on similarities between these SLP patterns, where high pressure seems a constant factor, and other (modern) droughts? Or to put it differently, is the circulation which led to the 1842 drought a unique one-off situation or do you see some similarities between the driver of this drought and other droughts? RESPONSE: Accepted, following changes in the manuscript were done as follows in the third paragraph of Section 3 Methods: "In order to investigate the synoptic patterns current in the driest months of 1842 (January, February, April–August), monthly SLP maps for the Atlantic-European area were created and also expressed as deviations from the SLP means in 1961–1990 (Fig. 6). Moreover, synoptic patterns of 1842 drought were compared also with synoptic patterns that typically occurred during other extremely dry months in Central Europe during the 1850–2010 period as defined by Brázdil and Trnka (2015). Using monthly resolution time series of Z-index and SPEI-1 of the above period, at first the 20% of the most severe droughts were selected. Only those months occurring in the selection according to both indices were denoted as extremely dry. Number of such extremes for each month from April to August in the 1850–2010 period varied from 27 (August) to 31 (April, May). Typical SLP fields and their variability were constructed as the mean (standard deviation) from all extremely dry months using the HadSLP2 database (Allan and Ansell, 2006). Consequently, SLP fields of 1842 drought were compared to the mean and variance of SLP fields of extremely dry months defined above and used now as a reference. Moreover, overall correlation between 1842 SLP fields and the mean SLP field of extremely dry months was calculated to evaluate the overall degree of similarity." The new second paragraph

was added to Section 4.2.: "Compared to other extreme droughts, that occurred over the territory of the Czech Lands, in 1842 only the April SLP field was quite exceptional because it shares only 10% of variability with the mean SLP of April extremes during the 1850–2010 period. The most significant feature in the April 1842 SLP field was a region of high pressure centred in the northern British Isles. Synoptic patterns of other months (May–August) in 1842 were more similar to the mean SLP of corresponding extremely dry months as they shared from 64% (August) to 90% (June) variability. Compared to the mean extreme SLP fields of 1850–2010, the 1842 SLP patterns do not show unambiguous pattern for all months. There occurred positive deviations in north-western Europe and negative deviations in north-eastern Europe especially in April, June and July while opposite dipole pattern occurred in May and August." Reference: Allan, R. and Ansell, T.: A new globally complete monthly historical gridded mean sea level pressure dataset (HadSLP2): 1850–2004, J. Climate, 19, 5816–5842, 2006.

*) page 9, line 17. Perhaps a little more information is needed here on the zero point. Is that also a supposed minimum as in an earlier example, or some other reference? RESPONSE: The corresponding sentence was changed as follows: "On the River Seine at Paris, the water level dropped for 52 days below the zero point on the Pont de la Tourelle scale, established already during an extraordinary drought in 1719 (Belgrand, 1872)." Reference: Belgrand: La Seine - Études hydrologiques - Régime de la pluie, des sources, des eaux courantes - Applications a l'agriculture, Dunod, Éditeur, Paris, 1872.

*) page 9, line 42-48. These islands in the southwest of the Netherlands are surrounded by sea water and have no rivers or freshwater lakes. This makes them vulnerable to drought. The situation is likely to be worse there than for other parts of the Netherlands. RESPONSE: Accepted. Based on the referee comment the paragraph was extended as follows: "Problems arising out of scarcity of water were especially acute in towns and cities. When a general and urgent lack of drinking water started

to occur in Middelburg, the capital of the province of Zeeland in The Netherlands, the town authorities found themselves obliged to send a ship to "mainland" Holland on 8 June in order to provide it (Nederlandsche Staatscourant, 11 June 1842). According to a report from the city of Goes on the island of Zuid-Beveland (The Netherlands) dated 13 June, good drinking water was so scarce that many people were using bad water, and adverse health consequences were feared (Vlissingsche Courant, 17 June 1842). Because flat islands in the southwest of the Netherlands are surrounded by sea water and tidal rivers and have no rivers or freshwater lakes, they are vulnerable to drought. Their situation can be worse than in other parts of the Netherlands. But also the Lord Mayors of other Dutch cities like Breda (Bredasche courant, 14 Aug. 1842) and The Hague (Dagblad van 's Gravenhage, 19 Aug. 1842) took measures to limit the use of water. A report that appeared on 31 August indicated ..."

*) page 11, line 19-26 and page 15, line 1-8. The relation to drought is not clear – there is only a claim in the newspaper. The drought and pests plague may be coincidental without the drought being a cause. Please present independent evidence that drought relates to the outbreaks of these worms/insects. RESPONSE: We do not claim that they are surely and exclusively the consequence of drought; however, there is a higher probability of occurrence in these cases, and therefore these invasions should be mentioned as potential (direct or indirect) consequence. Following the referee's comments, we add a new paragraph at the end of Section 5.1 as follows: "Reporting locusts or over-reproduction of some insects or small mammals (Section 4.3.2) in connection with drought does not mean that it is exclusively only the consequence of drought. However, there is a higher probability of occurrence in these cases. Usually the probability of the occurrence of pests (i.e. insects, mice and other small mammals) increases during droughts (e.g. Yihdego, 2018), although this relationship is not always consistent (e.g. Maxmen, 2013). Clear connection was found between hydrological extremes (including drought) and mice (or rodent) pests (e.g. Pech et al., 1999). Wasps become more active and lay much more eggs in drought and/or warm weather (Romo and Tylianakis, 2013). Drought is also a most important stimulating factor for locusts to change to a

highly social gregarious phase, which leads to mass behaviour and higher activity (e.g. Rogers and Ott, 2015)." References: Yihdego, Y.: Drought and pest management, in: Eslamian, S. and Eslamian, F. (Eds.): Management of Drought and Water Scarcity, CRC Press, Boca Raton, 2018. Pech, R. P., Hood, G. M., Singleton, G. R., Salmon, E., Forrester, R. I., and Brown, P. R.: Models for predicting plagues of house mice (Mus domesticus) in Australia, in: Singleton, G. R., Hinds, L. A., Leirs, H., and Zhang, Z. (Eds.): Ecologically-based Management of Rodent Pests, Australian Centre for International Agricultural Research, Canberra, 81–112, 1999. Rogers, M. S. and Ott, S. R.: Differential activation of serotonergic neurons during short- and long-term gregarization of desert locusts, Proc. R. Soc. B, 282, 20142062, doi: 10.1098/rspb.2014.20162, 2015. Maxmen, A.: The threat of insects to agriculture is set to increase as the planet warms. What action can we take to safeguard our crops?, Nature, 501, S15–S17, 2013. Romo, C. M. and Tylianakis, J. M.: Elevated temperature and drought interact to reduce parasitoid effectiveness in suppressing hosts, PLoS ONE, 8, e 58136, doi: 10.1371/journal.pone.0058136, 2013.

*) page 12, line 49-51. I'm confused what this report actually means, apparently it is salt water intrusion in the waterways which will make the water useless for irrigation or drinking. But for a period of half an hour? I'm not sure if that is even possible. RESPONSE: Accepted and corrected. Corresponding text was changed as follows to remove possible confusion: "On 15 June, people in Emden (Germany) gave voice to a host of complaints about a lack of water. The ports and waterways had to be flushed with seawater, so all the city canals in their neighbourhood had become salty and therefore useless for cattle or irrigating the vegetable gardens around the city (Algemeen Handelsblad, 26 June 1842)."

*) page 13-14, lines 43-23. I don't see why this is added to the text. It does not clarify the impact of the drought, it only gives information on the science of that period (which is not the topic of the paper). RESPONSE: This is a part of Section 4.3 entitled Impacts and human responses to the 1842 drought. It means that we are not

only interested in the direct impacts, but also in responses to which belonged related activities of religious institution which can be attributed to social responses (see lines 43-52 on page 13 and lines 1-7 on page 14). As a rather important part of social reflections, we also discuss the perception of 1842 drought because, for example, it had reflection in contemporary sources with some practical outcomes like use the term "the wine of the eclipses". We believe that complex studies of individual drought episodes cannot omit aspects of social responses and their perception. From these reasons we see these two paragraphs as important part of our findings and we would see not as a good solution to remove this part from the text. In addition to the previous expression, it should be mentioned that the perception of a natural extreme was a significant part of everyday life for traditional society, their beliefs influenced their everyday activities, and therefore it is an important direct consequence (with many indirect further consequences) of a weather-related extreme such as drought. The sole description of facts about a drought event, alone in itself, does not fully describe the response of society – unless we also include perceptions of the event. The listed cases are particularly interesting as they can provide good parallels to the societal response of individual drought extremes in earlier periods (e.g. Middle Ages) when often only the information regarding perception can be traced, and not any other information is available regarding the drought itself. Thus, we find it rather necessary to include perception issues (e.g. spiritual response, scientific perception etc.).

*) page 14, lines 36-43. This text requires a little more interpretation. The difference between the ranking based on SPI and SPEI is confusing otherwise. Clearly, the amount of precipitation was very low (making the SPI very negative), but the temperatures were not that high (especially in winter and spring), making the estimate you make for PET relatively low. This is the only reason why the SPEI gives such mundane values. You see that the scPDSI, which is based on a water balance (in contrast to the SPEI) does not give as much weight to the PET value as the SPEI (it calculates the actual evapotranspiration) and gives a drought ranking that is more in-line with the SPI value. RESPONSE: Accepted. The second paragraph in Section 5.1 was changed as

follows: "In the context of instrumental series, April–September 1842 was the most extreme drought in the Czech Lands in the 1805–2012 period according to SPI-1, while it was the third worst in terms of Z-index and the fourth most severe according to SPEI-1 (Brázdil and Trnka, 2015). Drought chronology created by a combination of documentary and instrumental data indicates that, for JJA as well as for the summer half-year (April–September), the drought of 1842 had a return period of 200 years for SPI and 50 years for SPEI and Z-index (Brázdil et al., 2019a). The reason for difference between the 1842 drought rankings based on SPI on the one hand and SPEI with Z-index on the other can be easily explained based on Fig. 4. While precipitation anomalies are present at almost each station (and are reflected in very low SPI values), above average temperatures, which would result in higher potential evapotrasnpiration and therefore drive SPEI and Z-index values to negative values, are occurring only during some months. As a result, SPEI and Z-index rankings do not reflect the same gravity of drought compared to SPI. The summer of 1842 ..."

*) page 15, line 19-26. here additional interpretation is needed as well, why do these trees not pick-up the drought signal? Are they living is conditions which are temperature stressed rather than moisture stressed? RESPONSE: As follows from the text, trees applied in cited papers were used for reconstruction of hydroclimatic patterns, i.e. they were moisture stressed. We just mentioned here the fact that 1842 did not appeared among the extremely dry years in each of quoted papers. We are very sorry, but we cannot speculate what was reason for it being not involved in all details of corresponding dendroclimatological analyses. This question is out of the scope of our paper and should be addressed on leading authors of reported papers.

*) page 16, line 25-41. The increase in the slaughter of cattle because farmers are unable to provide fodder or water, must have decreased the price of meat - any clues for that? RESPONSE: We did not find such a case in the sources, and, in general, this is not necessarily true. Generally, in times of fodder shortage, by the time the animals got slaughtered might be after a longer period of starvation, and their meat

might mean or not a significant additional quantity. On the other hand, price increase of base food products (especially crops) usually also had an impact on other beverages, and meat prices might also rise or at least did not change significantly. Different is usually the case in regions strongly relying on cattle import, which areas are usually more vulnerable. In this year we found no significant increase of beef prices.

Very very minor issues *) page 2, line 19. Would 'Shortage' be a more appropriate translation than 'Poverty'? RESPONSE: Accepted and changed as requested.

*) page 13, line 35: I would expect that it was NOT permitted to smoke a pipe.... RESPONSE: Accepted and corrected, this was a typing mistake.

*) caption fig 10, the year should be 1873 RESPONSE: Accepted and corrected.

*) caption fig. 4, These are Monthly variations (not annual) RESPONSE: We believe that "Annual variation" is in this context correct (exactly it is annual variation of monthly air temperatures and precipitation).

Squintu, A.A. et al. (2019a), doi:10.1002/joc.5874 Squintu, A.A. et al. (2019b), Building long and homogenous temperature series in ECA&D: blending of neighbour series, J. Applied Meteorology and Climatol. (submitted). RESPONSE: Both papers will be quoted in Section 2.2 and were included in References.

| 17 | Uccle | be | tg | 1833-01-01 | 2008-01-31 || 17 | Uccle | be | tn | 1833-01-02 | 2011-01-23 || 17 | Uccle | be | tx | 1833-01-06 | 2011-09-30 || 27 | Praha-Klementinum | cz | rr | 1804-05-01 | 2005-04-30 || 27 | Praha-Klementinum | cz | tg | 1775-01-01 | 2005-04-30 || 27 | Praha-Klementinum | cz | tn | 1775-01-01 | 2005-04-30 || 27 | Praha-Klementinum | cz | tx | 1775-01-01 | 2005-04-30 || 11744 | Dresden (Mitte) | de | rr | 1828-01-01 | 1992-11-30 || 11744 | Dresden (Mitte) | de | tg | 1828-01-01 | 1915-12-31 || 11744 | Dresden (Mitte) | de | tn | 1828-01-01 | 1915-12-31 || 11744 | Dresden (Mitte) | de | tx | 1828-01-01 | 1915-12-31 || 48 | Hohenpeissenberg | de | rr | 1781-01-01 | 2019-06-30 || 48 | Hohenpeissenberg | de | tg | 1781-01-01 | 2019-06-30 || 49 | Jena

Sternwarte | de | hu | 1824-11-20 | 2019-06-30 || 49 | Jena Sternwarte | de | rr | 1826-12-01 | 2019-06-30 | | 49 | Jena Sternwarte | de |tg | 1824-01-01 | 2019-06-30 || 49 | Jena Sternwarte | de | tn | 1824-01-01 | 2019-06-30 | | 49 | Jena Sternwarte | de| tx | 1824-01-01 | 2019-06-30 || 28 | HELSINKI KAISANIEMI | fi | tg | 1828-10-04 | 2001-12-31 | | 271 | Armagh | gb |hu | 1838-01-02 | 2008-12-31 | | 271 | Armagh | gb | rr | 1838-01-01 | 2001-12-31 || 257 | CET Central England | gb | tg | 1772-01-01 | 2003-05-31 || 274 | Radcliffe Meteorological Station Oxford | gb | rr | 1827-01-01 | 2019-04-30 || 274 | Radcliffe Meteorological Station Oxford | gb | tg | 1815-01-01 | 2019-04-30 || 274 | Radcliffe Meteorological Station Oxford | gb | tn | 1815-01-01 | 2019-04-30 || 274 | Radcliffe Meteorological Station Oxford | gb | tx | 1814-12-31 | 2019-04-29 || 169 | Bologna | it | rr | 1813-01-01 | 2007-12-31 || 169 | Bologna | it | tg | 1814-01-01 | 2003-12-31 || 169 | Bologna | it | tn | 1814-01-01 | 2003-12-31 || 169 | Bologna | it | tx | 1814-01-01 | 2003-12-31 || 171 | Genoa | it | rr | 1833-01-01 | 2008-12-31 || 172 | Mantova | it | rr | 1840-04-01 | 2008-12-31 || 173 | Milan | it | tg | 1763-01-01 | 2008-11-30 || 173 | Milan | it | tn | 1763-01-01 | 2008-11-30 || 173 | Milan | it | tx | 1763-01-01 | 2008-11-30 || 380 | Padova | it | tg | 1725-01-12 | 1997-05-31 || 380 | Padova | it | tn | 1774-01-01 | 1997-05-31 || 380 | Padova | it | tx | 1774-01-01 | 1997-05-31 || 381 | Palermo | it | rr | 1797-01-01 | 2008-12-31 || 85 | St. Petersburg | ru | tg | 1743-03-04 | 1999-08-31 || 10 | Stockholm | se | tg | 1756-03-27 | 2003-12-31 || 426 | Uppsala | se | rr | 1840-01-01 | 2001-12-31 || 426 | Uppsala | se | tg | 1840-01-01 | 2001-12-31 || 426 | Uppsala | se | tn | 1840-01-01 | 2001-12-31 | | 426 | Uppsala | se | tx | 1840-01-01 | 2001-12-31 RESPONSE: Many thanks for these detail information. All these series were considered on the base of monthly temperature and precipitation values, Prague-Klementinum also on the base of daily data.

---

## Author Comment (AC2) · 24 Aug 2019

The extreme drought of 1842 in Europe as described by both documentary data and instrumental measurements. This is overall a nice paper that sheds light on an important European drought event. The paper is well written and presented and does a good

job of illustrating the power of bringing together both qualitative and quantitative data to understand an important historical event. That said, I have some comments that need to be addressed by the authors. RESPONSE: We would like to thank the referee for generally positive evaluation of this manuscript. We are trying to respond below to all critical comments to contribute for further improvement of the paper.

The most significant comment I have is around the structure of the paper. It seems that results are presented throughout the paper including in the introduction which highlights some of the documentary impacts of the 1842 drought, and in the discussion where new results are presented to the reader. In particular I would interpret section 5.1 and 5.2 as results. I will leave it to the editor to decide this but I would prefer to see these integrated into results as the paper is using documentary and instrumental data. If the authors would prefer to have the proxy tree ring data as part of the discussion then I can understand that. The role of the discussion section should be to discuss the results and place them in a broader spatial/temporal context and to discuss any limitation, assumptions etc that were part of the analysis. The latter in particular could be fleshed out a bit more than is presently the case. Taken together with other comments below I feel that the outcome should be accept with minor revision as little new analysis would be required. RESPONSE: Concerning of the Introduction, the referee has probably in mind the paragraph on page 2, lines 15–29. In our feeling it is not presentation of the results, but some explanation and motivation which led us to study this particular drought event. From this reason we would like to preserve it how it is. Concerning of Sections 5.1 and 5.2, results of our research are clearly presented in Section 4, why here we put our results into context of finding from many other independent studies. So, in Section 5.1 we are confronting our results with findings from other papers based on instrumental and documentary meteorological data, i.e. here are not included any our direct results. Similar situation concerns also Section 5.2, where our results were put into context of dendroclimatological analyses and other phenological results, taken again from many other papers than is our recent study. From these reasons we believe that both sections 5.1 and 5.2 are not presenting results of our recent research and

from this reason they clearly belong to the discussion part.

Other comments Acronyms are used in the abstract, while some like NAO are widely known others like CEZI, SPI, SPEI, Z-index may not be, please spell these out for the reader. RESPONSE: Accepted and corrected.

It would be useful to have a map of Europe showing from where instrumental and documentary sources are derived from. This would help convey the continental nature of this event and its impacts. RESPONSE: Accepted, Section 2.2 was complemented by location of meteorological series used in graphic presentation. The maps indicating localisation of quoted documentary data have been prepared and are included in Supplementary material.

In describing the Pauling et al data please give a references for the gridded analysis from 1901-2000. RESPONSE: Accepted. On page 5, line 33, following reference was added: "Mitchell and Jones (2005)" Reference: Mitchell, T.D. and Jones, P.D.: An improved method of constructing a database of monthly climate observations and associated high-resolution grids, Int. J. Climatol., 25, 693–712, doi: 10.1002/joc.1181, 2005.

Does the Pauling et al data include precipitation from the individual series that you present later from across Europe and if so does this introduce a circularity into using these as independent pieces of information to assess the magnitude of the drought? RESPONSE: Pauling et al. (2006) used several long instrumental precipitation series from Paris (Slonosky, 2002), Kew (Wales-Smith, 1971), Bern (Gimmi et al., 2005), England and Wales (Wigley et al., 1984), Padova (Camuffo, 1984) and Estonia (Tarand, 1993) as predictors (among numerous other data types). It means that randomly selected series in Figs. 4 and 5 like Paris and Edinburg were used also in Pauling et al. (2006) in their spatiotemporal reconstruction of precipitation in Europe as presented in Fig. 2. However, we are using Pauling reconstruction (Fig. 2) and long precipitation series from several European stations (Figs. 4 and 5) just to demonstrate two different

aspects of 1842 drought: 1) its spatial distribution, and 2) its annual course of precipitation. We do not combine data used for these two aspects to derive any new (and potentially dependent) product. We do believe that our results and interpretations are not influenced by the fact that some analyses uses partly overlapping data.

In addition, perhaps I missed it but in the data section an overview of the precipitation gauges used later in the paper is not provided. In addition are there other series and regional precipitation records that might be usefully used to extend the quantitative assessment? RESPONSE: Station/regions represented in Figs. 4 and 5 were selected to keep any acceptable number of graphical examples and express well some territorial coverage over the territory of Europe. They were selected from the set of 33 temperature and 53 precipitation series going before 1842 and extending at least to 1990 (with respect to the 1961–1990 reference period), for which all analyses presented in both figures were provided too (for temperature only those stations with parallel precipitation measurements were considered). We used all data to which we had access and we could freely use them.

Why did you use SPI/SPEI 1 and not longer accumulations given that much of the focus is on agricultural, hydrological and socio-economic drought. Some thoughts on this either in the methods or in the discussion would be welcome. Does the result change if you do? RESPONSE: We used the drought indices and their 1-month versions to show contribution of individual months to the 1842 drought. Additionally we calculated for all stations/regions used also SPI and SPEI values for 3, 6 and 9 months. For stations/regions used in Fig. 5 they are presented as supplementary material to this article and at the end of the corresponding paragraph following sentences were added: "In order to present accumulated effects of drought, SPI and SPEI values for 3, 6 and 9 months were also calculated (see Figs. S2 and S3 in Supplementary material). Compared to Fig. 5, features of drought are well preserved but annual variation of indices is more smoothed a partly moved to the second half of the year demonstrating some persistence of drought patterns."

[Figure]

More detail is needed on how the drought indicators were derived, just saying that 'These series were then worked up' does not allow the work to be repeated. RE-SPONSE: The SPI, Z-index and SPEI were calculated using the methods provided on the page 6, line 26. However, it has been modified as follows: "These series were then worked up to calculate monthly drought indices: (i) Standardised Precipitation Index for one month – SPI-1 (McKee et al., 1993); (ii) Standardised Precipitation 25 Evapotranspiration Index for one month – SPEI-1 (Vicente-Serrano et al., 2010); and (iii) Z-index (Palmer, 1965) (Fig. 5). In case of SPEI-1 and Z-index, the calculation used the Thornthwaite method to estimate the potential evapotranspiration and data from the 1901–2000 period were used as reference period for all three drought indices. The SPI-1 and SPEI-1 were further complemented by their calculations for 3, 6 and 9 months."

Has the homogeneity of the various instrumental records used been assessed? If so/not this needs to be stated and if necessary returned to in the discussion. This is an early period in the observational history and gauges and their exposure often very different that today. More comment is needed on this. RESPONSE: Where it was possible, homogenised series of individual stations or regions were used, but it was not technically possible to homogenise other available non-homogenous series (due to missing reference series and metadata). Information of this type was added as the new second paragraph with figure of station locations to Section 2.2 as follows: "Using meteorological data from above cited and other sources, only stations including the 1842 year and ending not earlier than in 1990 (to have 1961–1990 reference period) were considered. This concerned finally 33 series of monthly temperatures and 53 series of monthly precipitation totals. But only ten of them extending over the studied part of Europe and having both temperature (T) and precipitation (P) data were used for graphic presentation (Fig. X – Selected European stations and regions used for graphic presentation of temperature, precipitation and drought indices in 1842). These series were from greater part homogeneous and some of them at least quality checked (Paris – T, Stockholm – P, Cracow – T,P)."

In terms of the consideration of hydrological drought why not look at 1842 in the context of the long term mean as you have done with precipitation? Only two adjacent years are used. Is the data not available? It seems it is from what is presented. RESPONSE: Compared to meteorological series, use of hydrologic series seems to be more problematic. Usually they are not homogenised and changes in catchments of corresponding rivers could mean quite significant breaks in homogeneity. From this reason we preferred only information from published papers or known sources (e.g. the Vltava or the Danube) with known quality of records or the use of extracted daily values from Wiener Zeitung for 1841–1843 to have at least some comparison with neighbour years. In other cases, as was for example daily data for the Meuse, only some part of this data was available. Moreover, the hydrologic data were used "only" for documentation of hydrological drought, complementing many documentary data reporting it.

Use of documentary sources is very good and indeed a standard to be aspired to. RESPONSE: Many thanks for this evaluation.

See points above on discussion where I think most work is needed in revising. RESPONSE: Please see our responses above and anticipated changes.

---

## Author Comment (AC3) · 24 Aug 2019

This is a historical climatological work of high quality based on an impressive amount of data and I recommend it to be published after minor revisions. RESPONSE: Many thanks for your recommendation.

[Figure]

In particular, it conforms to the current trend of combining different types of sources. Such source pluralism though, especially when the object of study belongs to modern times, entails rather superficial or non existent assessments of the quality of the sources, particularly the documentary evidence. Some words about the problems connected to such anthropogenic sources would be appropriate. The present disposition of the article also allows for that in a pedagogical sense; the documentary sources (2.1Documentary data) are presented first under 2 Data, and the not unequivocal reliability and representativity of these makes the following presentation of instrumental data the more relevant since the complementarity and usefulness of both source categories become clear. RESPONSE: Accepted, we extended the first paragraph of Section 3 reporting the work with documentary data as follows: "Documentary data extracted from the basic sources described in Sect. 2.1 were first critically evaluated with respect to the credibility of sources, their dating and the contents of the reports. Because of possible exaggeration of drought and its consequences and impacts (e.g. in newspapers), the content of every report was carefully evaluated with respect to described events and in case of doubts checked with other sources, also in temporal and spatial context. It concerned also some confrontation of documentary data with measured meteorological variables. If any such report was identified as doubtful, it was excluded from further analysis. In further checking of reports obsolete place-names were converted into their recent, more accessible form and the corresponding country added. Documentary data, by nature of a qualitative character, were employed, in particular, to descriptions of the character of hydrological, agricultural and socio-economic droughts from the point of view of environmental and societal impacts, as well as human responses."

The discussion about prices (pp 16f) is interesting but also highlights the problems connected to the use of prices to shed light upon climatic extremes. The authors are rightly cautious but the problems could be stressed even more. The climatological signal from agricultural prices is hard to detect without identifying and eliminating the 'noise' in the data and depends on whether market mechanisms are allowed to operate

freely, the degree of long-distance market integration etc, all of which is especially true for pre-industrial Europe when market demand for basic foodstuffs varied very little in the short run in contrast to supply. Some of these aspects are indeed mentioned in the present article and cannot be fully addressed there. But I would suggest that the clearly more significant importance of agricultural yields over agricultural prices should be recognized. RESPONSE: The referee is essentially right, but for us prices have no huge relevance, exactly because it is clear that the "noise" strongly smoothens the effects of bad harvests on prices. To explain all "noises" behind in such a broad region is the task which could be done properly only by particular economic historians, specialised on this period and dealing only with one country or region because of the complexity of the causative reasoning behind prices in this late period. Only very few economic historians dare to make Central European or other supra-regional analysis or conclusions. And since no such detailed economic historical analysis of the period is available, it is rather difficult to see through all possible reasons and mechanisms that influenced prices in all these regions in this detail. In response, at least, partly to this comment, the fourth paragraph in Section 5.3 was complemented as follows: "The fact that no particularly outstanding price increases for base products occurred in either Prague or Vienna could result from a number of factors and administrative strategies or decisions. One important influence was the mass storage of grain in royal/regional granaries; such a reserve could buffer the effects of at least one bad harvest (e.g. Jelenkor, 3 Sep. 1842). Similar to Hungary, this factor was present at that time also in other discussed countries (e.g. France, Germany, Czech Lands, Austria) which had, for example, granaries and an intensifying trade of basic agricultural products. It was also significant that the official market prices of major food products (meat, cereals, etc.) were fixed by regional authorities."

Finally, some small linguistic details: Page 2, line 3: "droughts events" should be replaced by "drought events" RESPONSE: Accepted and corrected.

Page 8, line 35: "positives modes" should be replaced by "positive modes" RE-

SPONSE: Accepted and corrected.

Page 13, line 1: "led a critical situation" should be replaced by "led to a critical situation"
RESPONSE: Accepted and corrected.

Page 13, line 35: "permitted" should be replaced by "prohibited" (or something similar)
RESPONSE: Accepted. We add "not permitted".

---

## Editor Comment (EC1) · Stefan Grab (Editor) · 10 Sep 2019

Dear Authors I am satisfied that necessary changes, as suggested by the referees, have been made and that the paper is now publishable. Please see the attached file with a few small editorial changes indicated. The paper will require further language editing but will leave that to the editorial office.

Sincerely Stefan Grab

[Figure]

The extreme drought of 1842 in Europe as described by both documentary data and instrumental measurements

Rudolf Brázdil[1], Gaston R. Demarée[3], Andrea Kiss[4,5], Petr Dobrovolný[1], Kateřina Chromá[2],
Miroslav Trnka[2,6], Lukáš Dolák[1], Ladislava Řezníčková[1,2], Pavel Zahradníček[2,7], Danuta
Limanowka[8], Sylvie Jourdain[9]

[1]Institute of Geography, Masaryk University, Brno, Czech Republic
[2]Global Change Research Institute, Czech Academy of Sciences, Brno, Czech Republic
[3]Royal Meteorological Institute of Belgium, Brussels, Belgium
[4]Institute for Hydraulic Engineering and Water Resources Management, Vienna University of
Technology, Vienna, Austria
[5]Department of Historical Auxiliary Sciences, Institute of History, University of Szeged, Hungary
[6]Department of Agrosystems and Bioclimatology, Mendel University in Brno, Brno, Czech
Republic
[7]Czech Hydrometeorological Institute, Brno, Czech Republic
[8]Rydla 17, Kraków, Poland
[9]Météo-France, Direction de la Climatologie et des Services Climatiques, France

*Correspondence to*: Rudolf Brázdil (brazdil@sci.muni.cz)

**Abstract.** Extreme droughts are weather phenomena of considerable importance, involving significant environmental and societal impacts. While those that have occurred in the comparatively recent period of instrumental measurement are identified and dated on the basis of systematic, machine-standardised meteorological and hydrological observations, droughts that took place in the pre-instrumental period are usually described only through the medium of documentary evidence. The extreme drought of 1842 in Europe presents a case in which information from documentary data can be combined with systematic instrumental observations. Seasonal, gridded European precipitation totals are used herein to describe general DJF, MAM and JJA precipitation patterns. Annual variations in monthly temperatures and precipitation at individual stations are expressed with respect to a 1961–1990 reference period, supplemented by calculation of selected drought indices (Standardised Precipitation Index SPI, Standardised Precipitation Evapotranspiration Index SPEI and Z-index). The mean circulation patterns during the driest months are elucidated by means of SLP maps, NAO and CEZI indices. Generally drier patterns in 1842 prevailed in January–February and at various intensities between April and August. The driest patterns in 1842 occurred in a broad zonal belt extending from France to eastern central Europe. A range of documentary data is used to describe the peculiarities of agricultural, hydrological and socio-economic droughts, with particular attention to environmental and societal impacts and human responses to them. Although overall grain yields were not very strongly influenced, a particularly bad hay harvest, no aftermath (hay from a second cut), and low potato yields led to severe problems, especially for those who raised cattle. Finally, the 1842 drought is discussed in terms of long-term drought variability, European tree-ring-based scPDSI reconstruction, and the broader context of societal impacts.

**1 Introduction**

Dry events, generally caused by reductions in precipitation totals compared to normal climatic conditions in a given area (meteorological drought), do not usually have such immediate and dramatic consequences (e.g. immediate loss of human lives, material damage) as might result from other hydrometeorological extremes – torrential rain, hailstorms, windstorms, floods, etc. The impacts of droughts appear over time, with some delay in the case of meteorological drought and progressively in agriculture (agricultural drought), water resources (hydrological and underground water drought), and society (socio-economic drought) (Heim, 2002; Mishra and Singh, 2010;

**Fig. 1.**

---

## Author Comment (AC4) · 12 Sep 2019

Dear editor, we would like to infor you that we accepted in our manuscript all English edits you proposed. With many thanks for your corrections and best regards Rudolf Brázdil in the name of all co-authors